# Nanotechnological Manipulation of Nutraceuticals and Phytochemicals for Healthy Purposes: Established Advantages vs. Still Undefined Risks

**DOI:** 10.3390/polym13142262

**Published:** 2021-07-09

**Authors:** Silvana Alfei, Anna Maria Schito, Guendalina Zuccari

**Affiliations:** 1Department of Pharmacy, University of Genoa, Viale Cembrano, 16148 Genoa, Italy; zuccari@difar.unige.it; 2Department of Surgical Sciences and Integrated Diagnostics (DISC), University of Genoa, Viale Benedetto XV 6, I-16132 Genoa, Italy; amschito@unige.it

**Keywords:** nanotechnology application, bioactive constituents of food, nutraceuticals, phytochemicals, food-grade NPs, food healthy properties, poor water solubility, possible NPs migration in food, toxicological risks of NPs ingestion

## Abstract

Numerous foods, plants, and their bioactive constituents (BACs), named nutraceuticals and phytochemicals by experts, have shown many beneficial effects including antifungal, antiviral, anti-inflammatory, antibacterial, antiulcer, anti-cholesterol, hypoglycemic, immunomodulatory, and antioxidant activities. Producers, consumers, and the market of food- and plant-related compounds are increasingly attracted by health-promoting foods and plants, thus requiring a wider and more fruitful exploitation of the healthy properties of their BACs. The demand for new BACs and for the development of novel functional foods and BACs-based food additives is pressing from various sectors. Unfortunately, low stability, poor water solubility, opsonization, and fast metabolism in vivo hinder the effective exploitation of the potential of BACs. To overcome these issues, researchers have engineered nanomaterials, obtaining food-grade delivery systems, and edible food- and plant-related nanoparticles (NPs) acting as color, flavor, and preservative additives and natural therapeutics. Here, we have reviewed the nanotechnological transformations of several BACs implemented to increase their bioavailability, to mask any unpleasant taste and flavors, to be included as active ingredients in food or food packaging, to improve food appearance, quality, and resistance to deterioration due to storage. The pending issue regarding the possible toxic effect of NPs, whose knowledge is still limited, has also been discussed.

## 1. Introduction

Consumers’ preferences toward health foods, as well as awareness of the existence of phytochemicals in plants, with health properties, are increasingly expanding and encompass both the food market and the natural compounds sector. Presently, the desire to make the most of these beneficial compounds as part of a normal diet, as food supplements (FS), or even as safer natural therapies, is strong and ever-growing. Currently, in Western societies, the food and plant derivatives industry, in addition to requiring the development of new functional foods (FF) and FS, is progressively seeking new nutraceuticals and phytochemicals. A further use of BACs contained in edible products is their incorporation into conventional foods and food packaging (FP) as preservatives, instead of using synthetic additives to improve food quality (such as shelf life, taste, and appearance). Furthermore, the development of new functionalities, particularly related to health promotion, such as antioxidant, anti-free radicals, and anticancer abilities, is highly desired by manufacturing companies and is the subject of extensive investigation by food scientists, food engineers, and food technologists. Most food-related BACs have been isolated from vegetables, herbs, fruits, legumes, oils, spices, nuts, and whole grains and like several phytochemicals have shown numerous beneficial effects, including antifungal, antiviral, anti-inflammatory, antibacterial, antiulcer, anti-cholesterol, hypoglycemic, immunomodulatory, and antioxidant activities [1,2,3,4,5]. Although the idea of exploiting the strong potential of foods, nutraceuticals, and/or phytochemicals for health purposes is brilliant, its realization is far from simple [6]. First, it must be considered that to have an effective contribution to reducing the risk of some diseases, a conspicuous daily dietary intake of BAC is essential.

In this regard, note that industrially processed foods contain fewer healthy compounds than fresh or frozen foods, due to degradation reactions that can occur during extraction or thermal decomposition procedures that can take place during heat treatments such as cooking [7,8,9]. To address this problem, an interesting solution for increasing the concentration of bioactive compounds could be to incorporate isolated BACs into commercial food products, obtaining BAC-enriched FF. An alternative strategy for pursuing the daily ingestion of an effective dose of BACs could be to formulate BACs in pharmaceutical-like preparations, such as FSs, which can be taken in addition to the normal diet. Unfortunately, the poor stability of BACs and the low solubility rate severely limit the feasibility of both resolutions [10,11,12]. Furthermore, the possibility of exploiting extracted BACs as effective orally administrable promoters of health is almost utopic. Note that for any bioactive molecule, solubility in an aqueous environment, permeability across the membrane of epithelial cells, and molecular interactions in the gastrointestinal tract (GIT) fluids are pivotal factors that strongly influence its path to the bloodstream, its final distribution to the targets, and therefore its effectiveness. Unfortunately, most BACs are insoluble in water and, in whatever form they are introduced orally, if not properly manufactured, they rarely reach the site of action in therapeutic concentrations. Moreover, degradation by chemical digestion, both enzymatic and microbial, which occurs in the mouth, stomach, and small and large intestine [6], further reduces the therapeutic effects possessed by BACs. Furthermore, even if a small fraction of BACs and their active metabolites manage to reach the bloodstream, they frequently undergo fast metabolism and are rapidly excreted via the kidney, biliary, or lung [13]. Collectively, BACs achieve only nano/picomolar concentrations in cells and tissues, which are insignificant doses for producing a health promotion response. Figure 1 exhibits the main criticisms and events that limit the beneficial effects of BACs after oral administration.

To overcome the aforementioned problems and exploit nutraceuticals and phytochemicals as health enhancers, researchers engineered several nanomaterials and resorted to nanotechnology and nanostructures with dimensions of nanometers (nm). To overcome the poor solubility, permeability, and negative pharmacokinetics of BACs, different nanosized delivery systems have been developed [6]. Additionally, food- and plant-related edible NPs have been and are designed and developed to act as color additives, flavorings, and preservatives, as well as to improve bioavailability, food shelf-life, taste, and appearance, using different techniques [14].

To give readers an idea of the growing scientific interest in the application of nanotechnologies in the food sector in the last 20 years, we carried out a survey of the number of works published year by year from 2000 to today, doing a search by keyword (nanomaterials, food) in Scopus (Figure 2).

While the interest in the topic was low among researchers in the years 2000–2010, in the last decade, it has grown considerably, and consequently, several papers have been published, especially very recently (2018–2021). This review provides an up-to-date overview of nanotechnological approaches for formulating the most relevant nutraceuticals and phytochemicals in more bioavailable food-grade NPs as part of medical treatments and/or to mask any unpleasant taste and flavors associated with isolated BACs. Furthermore, the use of BACs-enriched NPs as ingredients to develop active FFs, FSs, and FPs to improve the appearance, taste, quality, and spoilage resistance of foods, along with their storage, is also discussed. Finally, it should not be omitted that, despite the potential benefits of the application of nanotechnology, there may be a risk, not yet fully disclosed, for humans and the environment associated with an extensive application of NPs. We therefore also paid attention to the pending issue relating to the possible toxic effect of NPs on humans, animals, and the environment and their impact on customers’ acceptance, in a scenario of limited knowledge.

## 2. Nutraceuticals, Phytochemicals, Food, and Edible Plants: A Plethora of Benefits

BACs contained in edible products are compounds that are normally ingested with the diet and possess beneficial properties exploitable for enhancing humans’ health. They belong to two major categories: nutraceuticals, which are nutrients essential for human life and can derive from the animal, plant, or fungi kingdom, and phytochemicals, which are non-nutrients and derive from the plant kingdom [15]. When ingested and metabolized, several foods can provide numerous health benefits, and in vitro studies have demonstrated that many BACs are multifunctional compounds, which have healthy properties like those of conventional drugs (which in the case of nutraceuticals are added to their nutritional values) and can be considered “pharmaceutical-grade compounds”. A very dynamic sector extensively works for either obtaining from foods and plants the already known BACs, or for detecting, isolating, and characterizing ones that have never been reported. Optimized extraction methods [16,17] and suitable and conservative determination techniques are used to identify and quantify old, recent, and new BACs [18]. Based on their chemical structure and functional groups, BACs are classified into different families [19,20,21].

Table 1 collects the main families of BACs, and for each one, a brief description, the sources, the nutritional function (if present), and the reported health properties are included.

Vitamin C sodium ascorbate salt possesses a strong apoptotic activity on neuroblastoma (NB) cells, while ellagitannins (ETs) have shown activities against several degenerative diseases, such as cardiovascular, cancer, and central-nervous-system-disabling disorders, including Parkinson’s disease, Alzheimer, multiple sclerosis, and amyotrophic lateral sclerosis. The same effects were displayed by gallic and ellagic acids (GA and EA), which are ETs’ metabolites, in both in vitro studies and animal models [22,23,24,25,26,27]. Since it was established that most human diseases are produced by oxidative stress (OS) caused by an uncontrolled accumulation of reactive oxygen species (ROS) and reactive nitrogen species (RNS), the BACs benefit for human health mainly depends on their free radical scavenging activity, their capability to inhibit the assembly of microtubule and microfilament, to chelate metals and to inhibit protease [28,29,30]. In addition, GA, EA, and other natural products such as propolis extracts, encompassing phenolic acids, flavonoids, terpenes, and essential oils, can treat infectious diseases caused by bacteria belonging to several species, including those of sporogenic type. GA was proven to show antibacterial activity against *Paenibacillus larvae*, *Staphylococcus aureus* [31,32], *Escherichia coli*, *Pseudomonas fragi*, *Pseudomonas fluorescens*, *Pseudomonas putida*, *Pseudomonas* spp. P304, *Plesiomonas schigelloides*, and *Schigella flexneri B* [33,34,35]. 4-(2-Hydroxy-ethyl)-benzene-1,2-diol-, a polyphenol extracted from olive oil, tumerone and curcumin from *Curcuma longa* L. rhizome, GA, protocatechuic acid, quercetin, and resveratrol (RES), which are contained in numerous fruits and vegetables (Figure 3), are capable of inhibiting platelet aggregation and ROS activity induced by thrombin, collagen, or other agonists [15,36].

Allicin (from garlic), curcumin (from turmeric), catechins (from tea leaves), and RES (from grapes) (Figure 3) can help in preventing and treating cancer, cardiovascular illness, neuronal degenerative diseases, diabetes, and infections. Most of them are inhibitors of platelet aggregation and prevent OS triggered by ROS [15].

Terpenoids are one of the largest groups of natural products, mainly extracted from plants, which account for more than 40,000 compounds, but new compounds are discovered incessantly every year. They possess anticancer effects against several tumors, including breast, mammary, skin, lung, forestomach, colon, pancreatic, and prostate carcinomas. Additionally, most triterpenoids suppress cancer cells without exerting toxicity towards normal cells [37]. Terpenoids, which are found mainly in spices and di- and tri-terpenoids extracted from different typologies of *Salvia*, have shown antibacterial, hypoglycemic, and anticancer properties [4,5]. Presently, not only do BACs attract the food industry, but an increasing interest is arising also for their applications in cosmetics, as skin and hair care products, or in medicinal formulations, for enhancing their impact on beauty and health [15]. As BACs, whole foods and edible plants can also provide beneficial effects after ingestion.

Epidemiological studies have evidenced that red wine consumption is related to a reduction in mortality by cardiovascular events triggered by atherothrombosis, due to its capability of decreasing the progression of atherosclerotic lesions [15]. Green tea has proved to have protective effects on cardiovascular diseases [15].

## 3. Solubility of BACs

Low aqueous solubility affects most BACs; indeed, more than 40% of the natural and synthetic pharmaceutical molecules are practically water-insoluble [38]. GIT permeation of BACs is governed by solubility, which represents a pivotal requisite to achieve systemic drug concentration necessary for having a significant therapeutic response [39]. The dissolution of BACs in the intestinal fluids depends on their aqueous solubility and remarkably contributes to its bioavailability. Poor water solubility means a low absorption rate at gut level, low bioavailability, and insufficient blood and tissue concentration. Additionally, low solubility influences the feasibility of formulating BACs in emulsions for improving their water dispersibility, thus requiring very high concentrations of surfactants, which can promote unpleasant side effects such as GIT irritation [40,41]. To address these issues, nanotechnology extensively operates to improve BACs’ solubility, absorption, and bioavailability by developing efficient solubilizing methods at low manufacturing costs, which minimize or even avoid the use of harmful excipients, solvents, co-solvents, emulsifiers, or other additives. Nanotechnology aims to solve the conflict between the exciting and very promising results obtained evaluating BACs in vitro and those unsatisfying results obtained in in vivo studies.

### 3.1. Factors Affecting the Solubility of A Substance

Various factors, including temperature, pressure, the chemical structure of the compound, and physical properties affect the amount of solute that can be dissolved in a solvent. Table 2 collects the main factors that can influence the solubility of a compound and its dissolution rate with the due explanations.

### 3.2. Techniques and Approaches for Enhancing Solubility of BACs

As reported in Table 3, BACs solubility can be modified by two modalities:Particle Engineering Techniques (PETs)Formulation Approaches (FAs)

PETs aim to improve the solubility of BACs by rearranging poor soluble molecules in very small particles, with higher solubility, without recovering to additives, such as stabilizers or surfactants. Mechanical particle-size reduction, cryogenic particle engineering [47,48], nanoprecipitation, nanosuspension, supercritical fluid processing [47,49], freeze-drying (FD), and spray-freezing (SF) are the most common methods carried out. On the contrary, FAs try to increase the solubility of BACs by formulating them in colloidal dispersions, consisting of mixtures of water/oil phases, stabilizers, solvents/co-solvents, and other excipients. Then, using spray-drying (SD), milling, and other techniques, FAs lead either to solid formulations, lipid formulations, or amorphous formulations. Emulsion and double-emulsion solvent extraction, SD, freeze-drying, liquid anti-solvent, and solvent evaporation techniques are the most used techniques to produce BACs-loaded microparticles (MPs) and NPs [15]. Except for SD and FD, all these techniques have the disadvantages of retaining a lot of residual solvent and of letting low encapsulation efficiency (EE%). Moreover, except for FD, these techniques could trigger thermal degradations, are multi-stage processes, and in most cases, require a micronization step by air jet milling to obtain the desired particle size and size distribution. This step can lead to the production of cohesive, and electrostatically charged products with occasional crystallographic defects [15]. On the contrary, the supercritical fluid extraction method, although high in cost, is a single-step process that allows achieving micronized dry powders with controllable particle size, morphology, and crystallinity, with shorter operation time [15]. Furthermore, the residual solvent content can be monitored [15].

## 4. Nanotechnology, Nano-Formulations, and Nanomaterials

Nanotechnology, nano-formulation techniques, and nanomaterials are strongly involved in the current strategies developed to address the issues concerning BACs bioavailability. BACs solubility, delivery, and cell uptake can be improved by using NPs, as well as BACs protection from early degradation and fast metabolism can be realized by recovering to NPs. Moreover, by the nanotechnological manipulation of BACs is possible to prepare BACs-enriched food products (FFs) without interfering with the sensory and quality properties of the original food. The market for nanotechnological items produced in the food and beverage sector as health promoters are incessantly increasing. In the last 15 years, the global economic impact of nanotechnology has passed USD 1 billion to be USD 3 trillion by 2020, employing six million laborers worldwide [50]. A typical BAC engineered by nanotechnology is EA, a polyphenol found in fruits and vegetables, whose several healthy properties are unfortunately associated with very poor solubility and numerous pharmacokinetic drawbacks. Several studies exist on the adoption of appropriate NPs-based devices to enhance EA solubility, its hydrophilic-lipophilic balance (HLB), and GIT absorbability, as well as to protect it from early metabolism [14,51,52,53,54,55,56]. In particular, high water solubility was achieved using cyclodextrins [54,55], pectin, and polyester-based dendrimers [56].

### 4.1. NPs-Mediated Controlled Release

The controlled and targeted release of BACs from NPs has been identified as an essential stage of effective BACs administration. In vitro studies have established that a controlled delivery translates to a higher local concentration, while reducing the overall administered dose and consequently the systemic toxicity. pH, temperature, ultrasound or magnetic fields applications, light incidence, type and physicochemical features of NPs, chemical structure, physicochemical features of BACs, etc. are internal and external factors that can control the specific release of BACs [57].

BACs-loaded stimuli-sensitive nano-capsules possessing an oil core were shown to improve the BACs effects following the oral administration, due to targeted delivery and a controlled, long-term release, thus contributing to decrease the dosage and the administration frequency and to improve patient compliance [57]. The layer-by-layer self-assembly of pH-sensitive building blocks proved to be a promising approach to obtain biomaterials with customized properties, which were successfully applied as stimuli-responsive nanocarriers [15]. Starting from biocompatible pH-dependent polyelectrolyte, nontoxic nanocarriers with high permeability were designed [57].

In addition, BACs encapsulation in properly functionalized NPs can allow an increased cellular uptake and a slower BACs release, thus improving the BAC bioactivity, and contributing to a sustained therapy [15]. Rhodamine-loaded poly-alkylene glycol (PAG)-NPs were applied to SH-SY5Y NB cells or prostate cancer DU145 cells and were visualized by fluorescence. PAG-NPs were visualized in the cytoplasm, suggesting that they have been internalized via endocytosis, overcoming, without damage, the phospholipidic barrier of the cell membrane, which represents an impediment for hydrophilic compounds to enter the cells [58]. Furthermore, carrying BACs in NPs can favor BACs distribution in specific brain areas, thus providing more valuable benefits in neuro-regenerative treatments, while minimizing the BACs accumulation in the systemic circulation and related toxic side effects [59].

### 4.2. Nano-Formulations Techniques Based on Nanosuspension and Nanoemulsion Approaches

#### 4.2.1. Nanosuspension-Based Conventional Techniques

Nanosuspensions (NSs) technology is a technique suitable to improve solubility and bioavailability both of hydrophilic (H) and lipophilic (L) bioactive compounds (HBACs and LBACs). An NS is a colloidal dispersion of NPs in an aqueous media, stabilized by surfactants, co-surfactants, and polymers [15]. BACs NPs achieved by NSs techniques possess high dispersibility and solubility. They are exploitable for a sustained, controlled, and targeted delivery of the loaded BAC and possess improved stability and therapeutic effects in cells and tissues [60]. The conventional approaches to prepare NSs consist of the bottom-up and top-down methods (Figure 4).

The top-down techniques start from large-dimension particles and reduce their size to nanoscale dimensions by a media milling technique, high-pressure homogenization (HPH), ultra-high-pressure homogenization (UHPH), or supercritical fluid processes. The bottom-up methods start instead from BACs molecules and subject them to precipitation, melt emulsification, coacervation, inclusion complexation, or supercritical fluid extraction, thus causing self-association and self-organization, forming nanosized materials [15]. Table 4 summarizes the most common conventional top-down and bottom-up methods. The NS inclusion complexation technique was used for encapsulating β-carotene in *n*-octenyl succinate-modified starch, obtaining β-carotene-loaded NPs with improved dispersibility, coloring strength, and bioavailability [61]. Quercetin was instead subjected to HPH, achieving an NS of amorphous NPs [62]. Using SD technique and maltodextrin as a carrier agent, water-re-dispersible powders with radicals scavenging activity, reducing ability, and oxygen radical absorbance capacity, higher than those of the non-treated samples, were easily prepared. These powders proved to be suitable for the development of quercetin-enriched FF products [62]. A stable aqueous NPs (150 nm) suspension of α-tocopherol, with improved solubility and bioavailability was obtained by supercritical assisted process [63].

#### 4.2.2. Nanosuspension-Based Combined Techniques

To improve the performances of the single approaches, combination methods were born by merging the top-down and bottom-up techniques.

##### Nanoedge™ Technique

Nanoedge™ technique (Baxter Healthcare) consists of a microprecipitation step in water, using water-miscible solvents, such as methanol, ethanol, or isopropanol to dissolve the BACs (bottom-up phase). This action leads to an amorphous or partially amorphous precipitate. Subsequently, through HPH by piston-gap homogenizers, or through sonication, in a short period of time, nanosized particles (80–700 nm), endowed with great stability due to their crystalline physical state, are obtained (top-down phase). A slightly modified procedure, involving the complete removal of the solvents before HPH by evaporation, provides a solvent-free starting material to be subjected to HPH [15,68]. Performing a “Nanoedge-like” technique consisting of a microprecipitation followed by sonication, all-trans retinoic acid (ATRA) was formulated in 155-nm-sized particles, suitable for oral administration, in 30′ operation time [15].

##### H 69 Technology

H 69 Technology makes part of the SmartCrystal^®^ technology group and is like the NanoEdge™ approach. This technique differs from NanoEdge™ approach because the just-formed BACs particles obtained by microprecipitation are immediately treated with cavitation, particle collision, and shear forces. Practically, the precipitation (bottom-up phase), happens almost simultaneously to homogenization (top-down phase), allowing better control of the particle crystallization and stopping the nucleation, thus avoiding the crystals’ growth. Although this technique does not permit the removal of organic solvents, highly stable drug nanocrystals in the range of 20–900 nm can be obtained [15]. The H 69 process was performed to formulate RES in particles of 150 nm, suitable for oral administration, after 10 cycles HPH at 1200 bar [69].

##### H 42 Technology

H 42 Technology belongs to the platform of SmartCrystal^®^ technology as H 69 and mixes a bottom-up step, consisting of the BACs precipitation by SD performed in aqueous media containing surfactants, with an HPH phase (top-down phase), for further particle size reduction. Surfactants (poloxamers or sugars such as mannitol) are required to enhance the performances of the SD step. Short processing times, solvent-free dry intermediates, and small drug nanocrystals, obtainable after a reduced number of HPH cycles (170–600 nm) are the main advantages of adopting the H 42 approach. On the contrary, since high temperatures are necessary during SD, this technique is unsuitable for processing thermolabile compounds [15,70]. When H 42 was used for formulating RES, particles of 200 nm eligible for oral administration were obtained after only 1 HPH cycle at 1500 bar [15,70].

##### H 96 Technology

H 96 Technology (SmartCrystal^®^, Abbott/Soliqs, Ludwigshafen, Germany) consists of a bottom-up pre-treatment step involving the FD technique, which allows eliminating the organic solvents as in H 42, followed by the usual top-down step for particle size reduction through HPH. The frozen organic solutions are obtained using liquid nitrogen. Although the lyophilization step determines the extension of working times, H 96 technology is suitable particularly to process thermolabile or expensive drugs, due to the low temperatures and the high yields of the FD [15].

##### Combination Technology (CT)

CT is a process that, without using organic solvents, combines two top-down approaches. A low-energy pearl milling phase of an aqueous macro-suspensions that provide particles of 600–1500 nm, is associated with the usual HPH phase, thus obtaining particles with a size of 250–600 nm. The CT approach presents a limited risk of crystal growth, and the obtained NPs proved enhanced physical stability during storage. CT was used to formulate hesperidin, which is a poorly soluble glycosylated flavanone, present mainly within citrus fruits and especially abundant in the peel and pulp [15]. In vitro studies have established its antioxidant activity and when assumed with the diet, it has proven to be a valid vasoprotector. After five homogenization cycles at 1000 bar, hesperidin was obtained in the form of NPs with a mean particle size of 599 nm. Hesperidin NPs were characterized by improved solubility and long-term stability and were suitable both for oral administration and topical application [15]. Hesperidin nanocrystals can be found in the Platinum Rare cosmetic product (La Prairie, Volketswil, Switzerland). Furthermore, rutin and apigenin, which are very poorly soluble BACs belonging to the flavonoids family and possessing antioxidant properties were processed with the CT technology. By a pearl milling step followed by only one cycle of HPH at 100 bar, more soluble rutin NPs of about 600 nm, suitable both for oral and topical administration, were achieved. Apigenin NPs of 275 nm were obtained after one cycle HPH at 300 bar. Rutin nanocrystals are in a cosmetic product launched by Juvena, St. Margrethen, Switzerland [15]. In addition, to allow reducing processing times and costs, CT is suitable for scaling up [71].

#### 4.2.3. Emulsion-Based Techniques

Emulsions (EMs) technology is an approach suitable for reducing particles size of both HBACs and LBACs, to enhance their solubility and bioavailability and to obtain orally administrable drugs appropriate for pharmaceutical and functional food applications. Through EMs technology, BACs are encapsulated in small droplets mixing an aqueous phase (w) with an oil one (o) and obtaining water in oil (w/o), oil in water (o/w), or bi-continuous colloidal dispersions, which are stabilized using specific additives, such as generally-regarded-as-safe (GRAS) pharmaceutical surfactants, co-surfactants, and emulsifiers. Oils utilized in EMs comprise Captex 355, Captex 8000, Witepsol, Myritol 318, Isopropyl myristate, Capryol 90, Sefsol-218, triacetin, isopropyl myristate, castor oil, olive oil, etc. In w/o dispersions, aqueous droplets containing water-soluble HBACs are dispersed in an oil medium. In o/w dispersions, oil droplets containing LBACs are dispersed in an aqueous medium. Finally, in bi-continuous colloidal dispersions, microdomains of oil and water are inter-dispersed in the system. w/o EMs are adopted to formulate HBACs, while LBACs are usually formulated by o/w EMs. By EM techniques, emulsion-based delivery systems are obtained, which should protect the entrapped compounds and realize their controlled and sustained release at the target site [15,72]. Note that the use of charged surfactants can provide NPs with a net surface charge, with both positive and negative implications. Surface ionization could be useful for enhancing NPs’ capability to cross the biological membranes, thus allowing higher BACs concentrations to be achieved at the target site and higher therapeutic efficiency. On the contrary, an excessive number of charged groups on NPs surfaces could provoke severe damage to cells, causing permanent damage to their cytoplasmic membrane. Studies exist, which report that, while neutral NPs and low concentration anionic NPs can be utilized as colloidal carriers to deliver encapsulated BACs to the brain with good efficiency and trivial damage, cationic NPs have an immediate toxic effect at the BBB [15]. Based on the emulsion droplets’ size, EMs are classified as microemulsions (MEs) or nano-emulsions (NEs). Additionally, self-emulsifying drug delivery systems (SEDDSs) represent a third type of EM. Since MEs are not within the scope of the present review which aims at overviewing nanomaterials, we have not considered MEs in the present manuscript.

##### Nano-Emulsions (NEs) Preparation Methods

NEs are characterized by particles of 100–500 nm. While high-melting BACs are not suitable for NEs formulation technique, both HBACs and LBACs, as well as H/L food additives and H/L bioactive cosmetics, can be formulate using NEs. The obtained NPs are characterized by improved solubility, stability, and bioavailability, and an extended half-life. Although surfactants, co-surfactants, and stabilizers are required (5–10%), high drug loading (DL%) is possible, and solutions are isotropic, transparent, and kinetically stable [73,74]. NEs can be obtained either by low energy techniques, not involving mechanical devices, or by high energy techniques, needing the use of mechanical devices and strong agitation (Table 5). Low energy techniques use approaches known as “phase inversion temperature” and “phase transition” methods, associated with gently stirring. When low-energy techniques are used, usually the emulsion droplets formation occurs by self-assembling leading to larger drops, but also nano-dimensioned droplets can be achieved. Regardless of the techniques adopted, NEs stability is lower than that of MEs, because the very small droplets initially obtained, may tend to re-aggregate along time with the formation and growth of undesired great crystals. In high-energy techniques, droplets size depends on the type of stirring device used, on the operating conditions, and on the BACs physicochemical properties [75].

##### Self-Emulsifying Drug Delivery Systems (SEDDSs)

SEDDSs are suitable for orally delivering LBACs, food-grade chemicals additives, and, more in general, drugs. SEDDSs are anhydrous nano-dispersions achieved by drying a mixture of an oil phase, surfactants, co-surfactants/co-solvents, and LBACs through proper procedures, including SD or FD [65]. These powders will spontaneously arrange in colloidal EMs when merged with water or with GIT fluids [76]. SEDDSs include Self-Nanoemulsifying Drug Delivery Systems (SNEDDSs) (droplets size < 50 nm) and Self-Micro-emulsifying Drug Delivery Systems (SMEDDSs) (droplets size of 100–200 nm) [15]. SEDDSs can be taken orally by either solubilizing them in water and drinking the obtained NE or by ingesting capsules filled with gelatin (Figure 5).

**Table 5 polymers-13-02262-t005:** Some common methods for NEs preparation.

Methods	Components	Surfactants	Crucial Factors	Stirring	High Energy Mechanical Devices	Assembling
		Co-Surfactants				
		Co-Solvents				
Low energy	Oil phase Water phase Additives	Yes	BAC type Emulsion properties	Mild	No	Self
		Yes				
		Optional				
High energy	Oil phase Water phase Additives	Yes	BAC type Emulsion properties	High	Micro-fluidizers Ultra-sonicators High-Pressure Homogenizers	By energy devices
		Yes				
		Optional				
UHPH [77,78]	Oil phase Water phase Additives	Yes	BAC type Emulsion properties	High	Ultra-High-Pressure Homogenizers	By energy devices
		Yes				
		Optional				

When SEDDSs meet water or GIT fluids, by small agitation or by the digestive motility of the stomach and intestine, the nanosized oil in water droplets (o/w) form. Many formulation parameters, including the surfactants concentration, the oil/surfactant ratio, the polarity of the emulsion, and the droplets’ size and charge, determine the self-emulsification ability and influence the efficiency of oral absorption of LBACs from the SEDDSs. In addition, also the physicochemical properties of the BACs, such as pKa, log P, molecular structure, MW, presence, and quantity of ionizable groups have remarkable effects on the performances of SEDDSs.

Among the ideal candidates for SEDDSs development, there are those BACs with a low therapeutic dose. The self-double-emulsifying drug delivery systems (SDEDDSs), which can form water-in-oil-in-water (w/o/w) or oil-water-in-oil double EMs in GIT fluids, are novel self-emulsifying formulations, which represent a further evolution of conventional SEDDSs [79]. The main advantages of SEDDSs consist in a high oral bioavailability improvement and the possibility of an easy scale-up. SEDDSs allow high DL% and allow delivering peptides and lipids without the risk of lipid digestion. The group of Hu manufactured a SDEDDS loaded with epigallocatechin-3-gallate (EGCG), having improved photo-stability in respect of free EGCG [80].

##### Application of NE-Based Techniques in the Food Sector

NEs-based delivery systems have been exploited for formulating herbal drugs, whole plant extracts or their BACs, and food-related BACs, which are unstable in highly acidic pH and undergo liver metabolism if administered as free. Furthermore, using NE techniques, side effects due to BACs accumulation in the non-targeted areas can be minimized; therefore, the administration of such formulations is authorized also in pediatric and geriatric individuals [73].

NE techniques were used to nano-formulate turmeric, curcumin (diferuloylmethane), and di-benzoyl-methane (a structural analog of curcumin). Curcumin (E100, in the European code of food additives and C.I. 75,300 or Natural Yellow 3 in the cosmetic sector) is found in *Curcuma longa* and is used primarily as a coloring natural additive. Indeed, it has a yellow color like that of saffron. Curcumin is also used as GRAS FS, because possesses several health properties, including antiseptic, analgesic, antimalarial, and insect-repellent activities. Turmeric, in Southeast Asia, is commonly used to treat biliary disorders, jaundice, anorexia, cough, diabetic ulcers, liver disorders, rheumatism, inflammation, sinusitis, menstrual disorders, hematuria, and hemorrhage. Triacylglycerol was chosen as the oil phase and Tween-20 as an emulsifier to formulate curcumin in NE, which proved reduced toxicity, improved bioavailability and bioactivity and strong anti-inflammatory properties [15,81]. Other polyphenols of pharmacological interest such as tannins, stilbenes, and flavonoids, which have displayed in vitro antioxidant effects, have been encapsulated in NEs [82]. The in vivo poor antioxidant activity shown by EGCG was significantly increased when it was formulated in NE [83]. Bioactive lipids and carotenoids were formulated as NEs by observing, respectively, more stability against autoxidation and increased bio-accessibility. NEs were used to protect lactic acid bacteria and to restore the proper microbiota in diverse intestinal diseases conditions [84]. Table 6 summarizes some examples of emulsion-based devices prepared for delivering some BACs.

Ethyl acetate extracts of pomegranate peel, containing polyphenols, and rich in EA, were emulsified with pomegranate seed oil (as oil phase), achieving polyphenols-loaded NEs, suitable for topical applications. Upon application onto the skin, NEs allowed the mixture of polyphenols to permeate the skin barrier, thus reaching the epidermis and dermis and the deeper skin layers. Polyphenols-enriched NEs proved to act as an anti-photo-aging cosmetic, helping to avoid or delay UV radiation damage [92]. Lemongrass essential oil (LEO) is a volatile, hydrophobic mixture of unstable compounds extracted from the leaves and stalks of the lemongrass plant. LEO is often found in soaps and other personal care products as flavor, due to its citrus scent, but was traditionally used to treat digestive problems and high blood pressure. Presently, it is becoming popular as a tool in aromatherapy to relieve stress, anxiety, and depression. In addition, LEO possesses antimicrobial effects and is gaining particular interest and wide acceptance by consumers, due to its relative safety. Unfortunately, LEO is susceptible to autooxidation, thus degrading easily, losing activity, and providing smelly or even harmful compounds, responsible for allergic reactions and skin irritation. Finally, LEO application in food preparations or in FP industry, as an antimicrobial and preservative additive, is limited by the high volatility and by the strong odor of its constituents when in the free form.

NE formulation of LEO leads to the reduction of its undesired sensory impact while enhancing its antimicrobial activity. In particular, edible carnauba wax and LEO NEs were developed, achieving a coat packaging for protecting plums, which proved to inhibit the growth of food-borne *Salmonella* spp. and *E. coli* [93].

### 4.3. Nanomaterials for Formulating BACs

Drug Delivery (DD) is a specific engineered technology concerning the development of drugs carriers known as Drug Delivery Systems (DDSs). By using solid NP delivery systems (SNDSs), highly soluble bioactive nanomaterials can be obtained, either by physically entrapping or by covalently linking BACs. In addition to enhancing BACs solubility and bioavailability, the use of SNDSs improves BACSs dispersion and stability in GIT, enhances their systemic spread, permits better transportation through the endothelial cell layer, a controlled release at the target site, and allows the impact of the microbiota metabolism to be controlled, thus also improving BACs’ bio-efficacy [13]. Furthermore, the reduction of BACs particles size to nanometers, jointly with the presence of the encapsulating carrier, leads to the improvement of BACs cellular uptake, due to the decreasing of the repulsion factors and/or interfacial tensions [94,95]. Generally, once loaded in food-grade SNDSs, BACs that were originally water-insoluble become easily dispersible in aqueous media [96] and simultaneously become protected by the outer shell of the nanocarrier from oxidation, as well as from the acidic and alkaline degradation, which can occur in the stomach and small intestine, respectively [97,98]. For these reasons, BACs-loaded SNDSs can more easily reach the tissues affected by pathophysiological situations in concentrations adequate for exerting a therapeutic action. The SNDSs’ digestibility in the GIT or in others body districts determines the release of BACs. Therefore, to realize a targeted release of BACs, the materials for developing SNDSs should be selected on the basis of their physicochemical features, which should be suitable to permit SNDSs degradation where desired [13]. Additionally, note that the drug-loading method strongly influences the DL%, and in turn the final release of BACs, their bioavailability, and their bioactivity. Starch-based NPs are digested at an oral level by the activity of amylase, while polysaccharide- and protein/polysaccharide-based NPs are degraded in the small intestine, due to variations of pH and salt concentrations [98]. According to these degradative processes, the BACs release happens in the oral cavity from starch-based NPs, while in the small intestine from polysaccharide and protein/polysaccharide-based NPs. Furthermore, lipid-based NPs release BACs in the small intestine simultaneously with the digestion of triglycerides [13].

SNDSs with particles sizes in the range of 20–1000 nm can also positively influence the transport of BACs through enterocytes by transcellular endocytosis.

The surface charge of SNDSs could be responsible for the formation of hydrogen bonds with the mucosal surfaces, contributing to momentary retention [99], while the presence of surface cell-penetrating ligands could contribute to enhancing transmembrane transport efficiency [13], influencing positively the effectiveness and bioactivity of the transported BACs. Note that NPs equipped with a lipid phase can reach the bloodstream via mesenteric lymph and thoracic ducts, avoiding hepatic first-pass metabolism, thus extending the half-life of BACs-loaded SNDSs.

Efficiency, biocompatibility, and food-grade requisites are essential for SNDSs developed for oral administration and food-related uses. The main attributes that the SNDSs for food uses should possess are summarized in Table 7.

To prepare SNDSs for delivering BACs for health promotion purposes, natural materials have been mainly adopted. In any case, biocompatible and biodegradable synthetic polymers and copolymers such as polyethylene glycol (PEG), poly urethane (PUR), poly capro-lactone (PCL), poly lactic-co-glycolic acid (PLGA), polyvinyl alcohol (PVA) or poly 2-vinyl pyridine (P2VP) are also extensively under consideration. Micelles obtained with these polymers are present in many therapeutic devices approved by the Food and Drug Administration (FDA) [102]. In addition, synthetic biocompatible polyester-based dendrimers were synthetized for future evaluations, as nanodevices to dissolve, deliver, and protect BACs [103,104,105,106].

Table 8 reports some examples of nanosized formulations of BACs developed by using the abovementioned polymers.

Interestingly, the developed dendrimers [16,110,111,112,113,114] are characterized by a *core-shell* structure. When BACs were physically encapsulated, the resulting BAC-enriched dendrimers were characterized by having a food-related functional *core* and a dendrimer *shell*. On the contrary, when BACs were covalently bound, the dendrimer formulations were typified by a dendrimer *core* and a bioactive *shell*. The drug-loaded dendrimers showed a favorable drug release profile protracted in time and improved biological activities.

#### 4.3.1. The Main Types of Organic Biocompatible and Biodegradable Solid NPs (SNPs)

Currently, there are different types of SNPs, whose classification is based on their physicochemical nature, production method, properties, free energy, interactions type, and typology, etc. [115]. To date, the most adopted organic SNPs for BACs encapsulation include non-synthetic lipid-based NPs (LNPs), protein-based polymeric NPs (ProNPs), oligosaccharides-based NPs (ONPs), and polysaccharides-based NPs (PNPs). Inorganic metal oxide-based and clay-based NPs are also extensively used, but they are not within the scope of this review. Table 9 briefly summarizes the main properties and functions of organic solid NPs (SNPs) used to transport and deliver BACs.

##### Lipid-Based Nanoparticles (LNPs) 

Lipid-based NPs include solid-lipid nanoparticles (SLNPs), liposomes (LPs), micelles (MICs) (including normal micelles (n-MICs) and inverse micelles (i-MICs)), and niosomes (NIOs) (Figure 6) [121].

Solid-Lipid Nanoparticles (SLNPs)

SLNPs are emerging products of lipid nanotechnology [122,123], and in the last years, they have become the most popular and commercially available NPs for delivering LBACs, due to their natural composition and biocompatibility. SLNPs have a spherical morphology of 10–1000 nm, as detected by Transmission Electron Microscopy (SEM). SLNPs have an external lipid monolayer with a solid-lipid *core* able to solubilize lipophilic molecules (LBACs). The lipid *core* is stabilized by surfactants and emulsifiers, whose properties can influence SLNPs stability and size. The lipids used to obtain SLNPs include triglycerides (tristearin), diglycerides (glycerol bahenate), monoglycerides (glycerol monostearate), fatty acids (stearic acid), steroid molecules (cholesterol), and waxes (cetyl palmitate) [15]. As stabilizers, all classes of emulsifiers have been adopted to stabilize the lipid dispersion. It has been found that the combination of more emulsifiers can prevent drug expulsion due to its crystallization during storage [124], which is one of the crucial drawbacks related to SNPs. The percentage fat/aqueous medium ratio of 0.1:30.0 (*w*/*w*) is considered the best choice to develop SLNPs [125].

Liposomes (LPs)

LPs are nano-scaled artificial vesicles obtained by mixing phospholipids and cholesterol. LPs are considered of significant interest as immunological adjuvants and drug carriers [15] and consist of a lipid bilayer enclosing an aqueous *core* (Figure 6). They allow a high EE% towards BACs with a wide range of polarities, thanks to the possibility of including them either into the aqueous *core* of the phospholipid vesicle or into the bilayer interface [15].

LPs can preserve the included BACs from enzymes activity and degrading agents. LPs deriving from natural lipids, are biodegradable, biologically inactive, non-antigenic, non-pyrogenic, and do not present intrinsic toxicity. Although LPs suffer from instability in plasma, sterically stabilized LPs have been developed [15].

Micelles (MICs)

MICs are very slim, spherical lipid particles with dimensions of 10–400 nm, which can form either in aqueous solutions or in oil medium. MICs formed in aqueous solution, as PEG-PLGA micelles are normal micelles (n-MICs). On the contrary, micelles that occur in oil solution as PLC-P2VP micelles formed in the oleic acid medium are inverse micelles (i-MICs). n-MIC can solubilize LBACs in the non-polar inner *core*, while i-MIC are capable to solubilize HBACs in the polar inner *core* (Figure 6). Micellar delivery systems allow the intravenous administration of hydrophobic drugs, without using solubilizing adjuvants frequently associated with toxic symptoms [15]. MICs have been exploited to improve the bioavailability and systemic residence time of several BACs. MICs can protect BACs from early inactivation and possess high DL% and good stability [126]. BACs release from MICs can be influenced by MIC intrinsic stability, BACs diffusion rate, the partition coefficient, the copolymers biodegradation rate, BACs concentration within the MICs, BACs MW, BACs physicochemical features, and BACs location within the MICs [15]. Nevertheless, BACs release can be voluntarily provoked at the target site by local stimuli, as variation in pH, temperature, or the application of ultrasounds or light.

Niosomes (NIOs)

NIOs are uncharged lipid-based lamellar nanostructures (Figure 6) that are formed by merging non-ionic surfactants, including alkyl or di-alkyl polyglycerol ethers and cholesterol [15]. NIOs are an option of LPs for enhancing oral bioavailability of compounds with limited absorption. NIOs are less toxic for cells due to their uncharged structure. NIOs are vesicles osmotically active and stable, acting as reservoir systems and providing controlled and sustained delivery of BACs.

Examples of LNPs Applications

LNPs have been used to entrap EOs to achieve EOs-loaded lipid NPs, which showed the capability of reaching different types of cells and improved activity. Similar results were achieved by the nanoencapsulation of ferulic acid, a hydroxycinnamic acid abundant in plants, and tocopherol, an essential nutrient belonging to the lipophilic vitamins, present in many foods. Both compounds possess antioxidant activities [127]. Fatty acid-based micelles were used to solubilize and transport plant oxylipins, phytoprostanes, and phytofurans, which were derived by the non-enzymatic oxidation of linolenic acid [13]. By mixing Span 60 and Tween 60, with 15% PEG 400 as a solvent, a dermal delivery system consisting of EA-loaded NIOs was prepared. It exhibited very high EE% and high efficacy in delivering EA to human epidermis and dermis [128].

##### Oligosaccharide-Based NPs (ONPs): Cyclodextrins

Cyclodextrins (CDs) are suitable to develop formulations for preparing FFs, FSs, innovative food-related therapeutics, and smart FP. CDs are commonly used as host molecules in the monomolecular inclusion complex technique. CDs are cyclic oligosaccharides consisting of six (α-cyclodextrin), seven (β-cyclodextrin), eight (γ-cyclodextrin), or more glucopyranose units, which are linked by α-(1, 4) bonds [15]. CDs are obtained through enzymatic degradation of amylose by the enzyme cyclodextrin glucosyl transferase. They have a truncated cone structure and can accommodate hydrophobic molecules inside their hydrophobic interior cavity. CDs’ outer side, due to the presence of several OH groups, forms a hydrophilic layer, which confers CDs high water solubility (Figure 7).

CDs can improve solubility and chemical stability and can limit the early degradation and metabolism of the guest molecules transported. Furthermore, CDs can modify unpleasant taste and flavor, as well as realize a controlled release of drugs [129]. Low doses of CDs are well tolerated by humans, but high doses may cause some adverse effects such as diarrhea and soft stools. β-CDs are currently mostly used as devices for drug delivery, due to their having a diameter that is suitable for loading several non-polar BACs [129]. Different methods are available to prepare the inclusion complexes (ICPXs) of LBACs using CDs [130] and details are reported in Table 10 [131].

β-CDs are extensively used as carriers for applications in the food, pharmaceutical, and cosmetic industries. Most studies assert the improvement of stability, life, and water solubility of flavonoids or other BACs from plants [132] (Table 11). Linoleic acid (LA) is an essential polyunsaturated ω-6 fatty acid, abundant in many nuts, fatty seeds, and their derived vegetable oils, which is crucial for health. LA is susceptible to oxidation, providing molecules involved in hyperalgesia and allodynia, which is a form of a very unpleasant inflammation. Using CDs, LA thermal stability and resistance against the environmental degradation factors were increased [15]. RES stability and solubility were improved, forming RES-β-CDs inclusion complexes [15]. Carotenoids were instead encapsulated into 2-hydroxypropyl (HP)-β-CD, succeeding in enhancing their water solubility [133]. Lycopene (Lyc) is a molecule belonging to the carotenoid group, possessing antioxidant properties, easily degraded by light, thermal energy, and chemical reactions, and it is not water-dispersible in GIT fluids. Water-dispersible nano-sized lycopene-β-CD complexes have been prepared using supercritical fluids process [134]. As observable in Table 11, several polyphenols-enriched nanomaterials have been obtained using CDs, including hesperidin-loaded HP-β-CD and RES-enriched maltosyl-β-CDs [135].

Interestingly, using β-CD and dimethyl carbonate (DMC) as cross-linker, EA nano-sponges (NSs) of 423 nm with low polydispersity index (0.409) and high zeta potential (−34 mV) were prepared, which showed a DL of about 69%, a dissolution efficiency of 50 μg/mL, and a controlled in vitro release of about 24 h. In in vivo studies, EA-NSs displayed an improved oral bioavailability of EA. Amino acids and hydrolyzed soy proteins are endowed with an unpleasant bitter taste that negatively affects their use in preparing FFs or FSs as beverages, which become acidic. According to a study by Linde et al. [15] and the judgment from a panel of trained tasters, the inclusion of amino acids and hydrolyzed soy proteins in α-CDs lead to the alteration of the bitter taste perception of amino acids and to the reduction of that of hydrolyzed soy proteins.

##### Polysaccharide-Based Nanoparticles (PNPs)

PNPs represent an efficient option to improve BACs solubility and to achieve BACs transport and controlled release, ideally at the target site [15]. Additionally, PNPs can improve BACs stability, allowing their processability and permitting them to develop BACs-enriched FFs [136] and natural additives capable of enhancing food shelf-life, quality, look, and sensory properties [137]. PNPs are prepared from natural hydrophilic polysaccharides such as alginate, chitosan, hyaluronic acid, pectin, and cellulose derivatives (hydroxyethyl cellulose and carboxymethylcellulose) and proper cross linkers or other substances inducing polymer–polymer interactions (Figure 8).

The preparation methods include covalent crosslinking, ionic crosslinking, polyelectrolyte complexation, and self-assembly [138]. Tripolyphosphate sodium salt (PPT) or ZnCl_2_ anhydrous are commonly used as cross-linkers. BACs can either be physically entrapped during NPs formation, covalently attached to the precursor materials, or absorbed into NPs after their preparation. PNPs are often freeze-dried in the presence of a suitable cryoprotectant or spray-dried into a microparticulate powder. With SD, non-suggestable for thermolabile BACs, nanomaterials suitable for preparing suspensions for oral administration can be prepared. With FD, a fine powder can be obtained that is also suitable for direct administration by inhalation or for compression into tablets [15]. Combinations of spray-freezing (SF) and FD techniques have been also described. Combination methods consist of the direct processing of frozen NPs achieved from the first SF step in a FD second step. PNPs have a high affinity to mucosal layers of the cells present in the respiratory tract and GIT, thus favoring higher permanence in these compartments and higher bio-efficiency. PNPs are classified as polyelectrolytes, comprising cationic, anionic, and neutral compounds, and non-polyelectrolytes, on the base of their intrinsic charge [15]. The polysaccharide coat of PNPs can both protect the encapsulated BACs against early degradation and interact with specific receptors in cells and tissues, thus promoting cellular uptake and site-specific controlled targeting [15]. The most used cationic polyelectrolyte is chitosan, which is made of repeated units of D-glucosamine. It is non-toxic, biodegradable, possesses a mucoadhesive nature, allows a controlled release of encapsulated agents and their prolonged residence time at the site of absorption [139]. Chitosan, thanks to its positive charge, can complex and solubilize negatively charged macromolecules without resorting to toxic solvents during preparation. Chitosan allows the administration of several BACs, also food-derived, for treating diseases in several body compartments such as nasal, oral, ocular, and dermal [140]. In addition, chitosan NPs can be modified by inserting ligands recognized by receptors on cells membranes for a more efficient delivery [141]. Anionic polyelectrolytes include alginate, heparin, pectin, and hyaluronic acid sodium salts. Alginate is biocompatible, biodegradable, non-antigenic, mucho-adhesive and is suitable for incorporating positively charged compounds [142,143]. Combinations of alginate and chitosan have been proposed as alternative systems for enhancing the circulation time of BACs. Pectin, is a hydrophilic food-compatible polymer, present in several fruits (fruits fiber) and often used in food industry as a thickener excipient in jam preparation. Pectin is regarded as a safe ingredient with no limit on the daily intake and possesses per se beneficial and prebiotic properties [144,145]. NPs based on hyaluronic acid sodium salt are highly soluble and stable in aqueous media, non-toxic, and non-immunogenic. Hyaluronic acid-based NPs are particularly efficient for targeted delivery of anticancer drugs, due to hyaluronic acid affinity for hyaluronan receptors, which are highly expressed in tumor cells. Neutral PNPs are made of uncharged polysaccharides, such as dextran, maltodextrin, pullulan, etc. [142]. Since they are uncharged, these materials manage to incorporate BACs, thanks to the presence of hydroxyl groups. Dextran was used to prepare delivery systems able to escape the reticuloendothelial system, thus possessing long systemic residence time, circulation permanence, and higher efficiency [142]. Maltodextrin is a water-soluble complex carbohydrate obtained through chemical hydrolysis of cereal starches (corn, oats, wheat, rice) or tubers (potatoes, tapioca). When orally administered, it has a neutral taste, is easily digestible, and is usually well tolerated. Maltodextrins are digested like glucose; hence they are massively used by the bodybuilding industry to increase the intake of carbohydrates in the diet without resorting to sugar.

Applications of PNPs in the Food Industry

Chitosan was used to encapsulate an olive leaf extract (OLE), achieving, after SD, OLE-loaded microspheres, with a loading of 27% in terms of polyphenol total content, and a smooth surface morphology [15]. Chitosan was employed to encapsulate GA obtaining chitosan-based NPs with anti-hemorrhagic effects and caffeic acid (CA), obtaining antioxidant food-grade NPs [146,147]. Yerba mate (*Ilex paraguariensis*) from the *Aquifoleaceae* family is a South American plant species, whose leaves and stems are used by the local population to prepare beneficial infusion beverages rich in alkaloids, such as xanthines and methylxanthines (caffeine and theobromine), saponins, and polyphenols. These beverages are known for their stimulating, anti-obesity, anti-hyperlipidemic, anti-inflammatory, and anti-oxidative actions. Collectively, Yerba mate has been shown to improve lipid metabolism, to counteract OS, to protect lipids and DNA from oxidative damage, and to ameliorate the metabolic unbalanced events associated with diabetes [148].

Yerba mate extracts, containing 62 mg of GA/g Yerba mate, have been encapsulated in both calcium alginate and calcium alginate-chitosan systems [15]. The DL% was higher in alginate beads (85%) than in chitosan-coated alginate beads (50%), while the Yerba mate release was faster from chitosan-alginate formulations than from the alginate ones.

Probiotics are beneficial microorganisms capable of improving intestinal microflora. Unfortunately, the survival of probiotic bacteria, especially in the human GIT, is trivial. A current strategy to overcome this issue consists of administering probiotic living cells equipped with a physical barrier against adverse environmental conditions. By alginate-starch encapsulation, probiotics delivery systems, with improved storage stability, shelf-life, and resistance to acidic pH were prepared. Using the supercritical carbon dioxide method, which allows avoiding the exposure of probiotic bacteria to water, oxygen, solvents, or heat, further improvements were observed [15]. Maltodextrins are widely used for encapsulating flavors and polyphenols. The ethanol extracts of black carrots, which contain high levels of anthocyanins, were spray-dried using maltodextrins as coating agents, achieving a powder with very high anthocyanins content [15]. Maltodextrins were also mixed with Arabic gum. Briefly, a mixture of maltodextrins (60%) and Arabic gum (40%) was used for encapsulating procyanidins from grape seeds, obtaining procyanidins-loaded NPs with EE% of 89%, remarkable antioxidant activity, and improved stability. Low methoxylated pectin (LMP) was exploited to improve the water solubility of EA by its physical entrapment in the pectin matrix. By SD, a solid microdispersion with a DL% of 22% (*w*/*w*) and a water solubility increased by thirty times was prepared [56]. The prepared EA formulation is suitable to be used as an FS able to give fine suspensions for the oral administration of EA and to be used as an ingredient to prepare functional beverages. In addition to being tasty, palatable, and well-tolerated, EA-enriched pomegranate juice containing the pectin-based EA microdispersion produced several beneficial properties when administered to mice [149]. Pectin from citrus fruit has been used to encapsulate both polyphenols and anthocyanins extracted from *Hibiscus sabdariffa* L. by using the SD technique. The achieved NPs merged the beneficial activities of pectin on cholesterol metabolism and colonic functions and the antioxidant properties of polyphenols, thus providing a novel nutraceutical product suitable for a variety of applications in functional food manufacturing [15]. Gliadin is the greatest fraction of gluten protein, which is a by-product that forms during wheat starch isolation and possesses interesting properties. Using liquid anti-solvent precipitation, gliadin was formulated in NPs, which were further coated with LMP and high methoxylated pectin (HMP), achieving PNPs with negative surface charge. The pectin coating stabilizes the gliadin NPs towards environmental stressing conditions (pH, ionic strength, and thermal treatment), thus providing stable functional ingredients suitable for being processed in the food industry as texture modifiers, lightening agents, or delivery systems [136].

##### Protein-Based Nanoparticles (ProNPs)

ProNPs can be prepared through methods including desolvation, coacervation, emulsification, nanoprecipitation, SD, NP albumin-bound technology, self-assembly, electro-spraying, salting out, and crosslinking. To obtain ProNPs, both animal proteins (gelatin, collagen, albumin, casein, and silk protein), and vegetable ones (zein, gliadin, and soy protein) are used. All encapsulation methods are based on the precipitation of the protein dissolved in a suitable solvent by adding a non-solvent or by changing the physicochemical parameters of the protein solution (pH, salinity, or temperature) [150]. De-solvating agents, like alcohol or acetone, are often added under stirring to the protein water solutions to promote the dehydration of the protein, thus causing the change of its conformation from stretched to coil. The crosslinking methods allow the stability of the original protein to increase and allow delivery systems to achieve to deliver BACs in a sustained mode. Different types of crosslinkers are available including chemical, ionic, thermal, and enzymatic. The most used are either an 8% glutaraldehyde aqueous solution or calcium phosphate (Figure 9).

The problem of removing completely the cross-linker, which can induce toxicity to biological systems, and the necessary washing step, which requires dialysis process, thus being time-consuming, are the most concerns regarding ProNPs. To address these issues, a crosslinking method based on γ-irradiation of ProNPs in phosphate buffer (pH = 7.2), in the absence and/or presence of ethanol and methanol at 30% and 40% (*v*/*v*), was investigated and introduced. The results showed that by controlling the irradiation dose, it is possible to modulate the crosslinking density and the particle size. This innovative method allows ProNPs to be produced in a one-step procedure and the crosslinking and sterilization of the NPs to simultaneously be obtained [151].

The advantages of using ProNPs as delivery systems for BACs are listed below [151,152]:Simple manufacturing;Absence of the requirement of emulsification performance;Compatibility with the high-pressure emulsification processes;High freeze–thaw stability [152]Loading capacity trackable by ultraviolet (UV)-spectrophotometry, fluorescence spectrophotometry, or high-performance liquid chromatography (HPLC);Abundance of proteins in nature;Suitability to being transformed;Absence of strong deleterious effects on the biological systems in which they are applied;Susceptibility to modifications, due to the occurrence of functional groups;Possibility to achieve the desired biodistribution;Biocompatibility;Capacity of carrying several molecules;Stability.

In addition, ProNPs offer the possibility to add hydrophilic polymers (PEG) for improving circulation residence time [15]. ProNPs can stabilize food-grade Pickering EMs [152], are suitable for applications in frozen food and for functional food formulations [152].

ProNPs could be classified on the basis of their origin (animal or plant proteins) and on the basis of the diverse types of benefits and constraints associated with proteins provenance. Toxicity and/or infections associated with their application are related only to animal proteins, while plant proteins, due to their hydrophobic characteristics, lack toxicity and have a lower economic cost [151].

Applications of ProNPs in Food Industry

By using animal gelatin (type A), EGCG was loaded by the layer-by-layer (LbL) assembly method, achieving an EGCG-loaded film material with EGCG content of 30% *w*/*w* and maintained antioxidant activity [15]. Gelatin was also used to bind GA, thus increasing its stability with a retained bioactivity [15]. Zein is a vegetable protein contained in the seeds of cereals and particularly in corn. By Zein, EA-loaded biodegradable hollow zein NPs were prepared to improve the oral delivery of EA [153]. The achieved NPs were made of a nano disperse *core* of EA/Na_2_CO_3_ with a shell of zein. These NPs showed a mean dimension of 72 nm, good stability, high DL% and a significantly improved permeation ability in vitro. The oral administration of EA-HZNPs was effective against inflammation in carrageenan-induced mouse paw edema model and showed better pharmacokinetic parameters.

Rennet-gelled protein and whey protein gel particles were adopted to entrap probiotics microorganisms to protect them from degradation and digestion in the stomach. Probiotics storage stability, shelf-life, and resistance to acidic gastric pH were improved. Supercritical carbon dioxide extraction method was performed to avoid the exposure of the probiotics to detrimental water, oxygen, solvents, or heat, thus improving the viability of the precious microorganisms [15].

#### 4.3.2. Organo-Synthetic Biodegradable Polymer Nanoparticles (OBP-NPs)

OBP-NPs are widely used to prepare BACs-loaded delivery systems due to their biodegradability, simple design, easy preparation, and good efficiency in delivering. OBP-NPs can load BACs either by physical interactions or by covalently binding by utilizing their several chemical functions. By adjusting the hydrophilic/hydrophobic balance (HLB) of polymers or copolymers, NPs characterized by various shapes and morphologies can be prepared. Figure 10 shows some of the most utilized biocompatible and biodegradable organic polymers approved by the U.S. FDA and European Medicines Agency (EMA).

They are suitable to deliver drugs in humans for the oral administration of nutraceuticals and phytochemicals and for producing food-grade smart nanocomposites for FP, able to preserve food quality, looks, and taste along with storage.

##### Polyethylene Glycol (PEG)

PEG is a polyether extensively applied both in industrial manufacturing and in medicine. PEG is known also as polyethylene oxide (PEO) or polyoxyethylene (POE), and its structure is commonly reported as H-(O-CH_2_-CH_2_)*_n_*-OH.

PEG is almost biologically inert, and it can be found as an ingredient in many pharmaceutical products, where it mainly acts as an excipient, because is unlikely to interact with the bioactive constituent of the formulation. Additionally, in drug formulations, PEG could act as a matrix, as a stabilizer and/or as preservative material. Cetomacrogol is a macromolecule belonging to the PEG family, which is the basis of many skin creams and personal lubricants. It is also present in the toothpastes as a dispersant since it manages to bind water and helps to keep xanthan gum uniformly distributed throughout the toothpaste. PEG derivatives, such as narrow-range ethoxylates, are often used as surfactants. PEG is used as a hydrophilic block in the synthesis of amphiphilic block copolymers, employable either to create polymersomes or to induce complete fusion in LPs reconstituted in vitro. For clarity, polymerosomes are biomimetic systems with a behavior like phospholipids. Structurally, they are polymeric NPs, which can function as drug “reservoir”, whose outer membrane consists of amphiphilic polymers. In the food sector, PEG is widely used as an anti-foaming agent, both in solid foods and in drinks. By acting as a steric barrier, PEG can protect the PEG-coated NPs from opsonization [154]. Collectively, PEG is eligible as solvent, co-solvent, container, surfactant, stabilizer, preservative additive, dispersant, lubricant, protecting agent, etc. for preparing food-grade formulations of BACs. By reacting PEG or other polyols with isocyanate derivatives, polyurethanes (PURs) can be prepared.

##### Polyurethane (PUR)

PUR is a synthetic alternate copolymer consisting of organic blocks jointed by carbamate links, traditionally synthetized as abovementioned. Currently, due to the growing interest in sustainable “green” products, polyols derived from vegetable oils or resulting from renewable sources are attracting a high amount of consideration for preparing PUR [155]. Alternatively, bio-based PURs can be prepared by reacting polyamines with cyclic carbonates [156]. PURs are chemically inert, and no exposure limits have been established in the U.S.A either by Occupational Safety and Health Administration (OSHA) or by American Conference of Governmental Industrial Hygienists (ACGIH).

PURs containing aromatic isocyanates possess chromophores able to interact with light, and therefore they are rising interest to manufacture nanomaterials for producing coatings to protect light-susceptible foods during storage. PUR packaging represents 4.6% of the total. Furthermore, PURs are also utilized for manufacturing nanocomposites for food contact applications, such as conveyor systems, hoses and tubing, chutes and chute liners, hoppers, gaskets, and seals. A guidance document on the use of PURs in food contact applications was published on 10 August 2016, by the American Chemistry Council (ACC) [157].

##### Poly-(Lactic-co-Glycolic Acid) (PLGA)

PLGA is a biodegradable polymer widely exploited for the development of food-grade NPs and applied as a biomaterial for the controlled delivery and release of BACs [158]. PLGA is subjected to rapid metabolism by physiological hydrolysis, providing lactic acid and glycolic acid, easily processed via the Krebs cycle, thus allowing low systemic toxicity. However, surface modifications of PLGA, using non-toxic and blood-compatible materials, is necessary to increase PLGA-based NPs residence time in blood circulation [159]. Combinations of PEG with hydrophobic biodegradable aliphatic polyesters provide hydrophilic non-toxic segments, which can be used to render PLGA resistant against opsonizing and phagocytosis, thus contributing to extending its life in the bloodstream and tissues [84].

##### Polycaprolactone (PCL)

PCL is a biodegradable polyester used in the production of special PURs with higher resistance to water, oil, solvents, and chlorine. PCLs are often used as additives for preparing resins with improved mechanical properties. PLC is compatible with several materials, with which it can be mixed without problems. PLC is often mixed with starch to lower its cost and to increase biodegradability. It can be added to polyvinyl chloride (PVC) as a polymeric plasticizer. PLCs applications include controlled drug releases, tissue engineering, bone scaffolds, packaging, compost bags, etc. A variety of BACs have been encapsulated within PCL beads for achieving a controlled release and target drug delivery [160].

##### Applications of OBP-NPs in the Food Industry

A topical ointment was prepared with PEG and 5% pomegranate rind extract, containing a high concentration of bioactive polyphenols including EA (13%). This formulation was studied concerning its release profile and skin-permeation capability. It was found that the polyphenol-rich formulation exhibited good physicochemical properties [161]. In tests concerning the wound-healing activities, free EA was less effective in inhibiting neutrophil infiltration and collagen augmentation in rat skin than the ointment releasing the same amount of EA [162]. However, both products applied topically exhibited anti-inflammatory effects in a mouse model of contact dermatitis [162]. OBP-NPs (150–300 nm) made of PLGA, chitosan, and PEG, were loaded with EA (up to 100 μM), achieving EA-loaded PLGA-chitosan-PEG NPs, that were able to potentiate apoptosis-mediated cell death in HepG2 human hepatoma cells [163]. It is known that the hydroxyl groups of anthocyanins can be easily oxidized into quinones, causing the reduction of their biological activity. PLGA-based NPs stabilized by PEG were used to encapsulate anthocyanins, obtaining anthocyanins-loaded biodegradable NPs that showed an EE% of 60%, improved stability, extended life, and a biphasic release profile in vitro. Indeed, an initial abrupt release was followed by a continued protracted slower supply. In vivo, they proved anti-inflammatory and anti-neurodegenerative capacities, preventing memory losses in estrogen-deficient rats, and showed a neuroprotective power against Alzheimer’s dementia [164,165,166]. In addition, anthocyanins-loaded NPs significantly reduced the levels of apoptotic proteins and the expression of various inflammatory markers, cytotoxic compounds, and proinflammatory cytokines [165]. Finally, anthocyanins formulated as NPs significantly upregulated endogenous antioxidant genes, thus helping in the prevention of OS with consequent attenuation of the clinical symptoms of the Alzheimer’s dementia and reduction of DNA damage to a higher extent than the native non-conjugated kind [166].

By using PCL, EA-loaded PLC-NPs (EA-NPs), which proved to have high EE% and DL%, were prepared through the emulsion–diffusion–evaporation technique. EA NPs were compared to free EA in an in vivo study on New Zealand white rabbits. The oral administration of EA-NPs produced an EA plasma concentration 3.6-fold higher than that produced administering free EA [160].

Adopting amino polyols from soybean oil, several structurally different eco-friendly soybean-oil-based cationic PURs were prepared to develop edible food coatings with antimicrobial properties. These edible NPs could represent an alternative food ingredient for preserving food from bacteria degradation and for extending its shelf life. The structural and hydroxyl functionalities of the different amino polyols influenced the NPs’ morphology, mechanical properties, thermal stability, and antibacterial activity. The PURs-based NPs showed good antibacterial effects toward a panel of bacterial pathogens including *Listeria monocytogenes* NADC 2045, *Salmonella typhimurium* ATCC 13311, and *S. minnesota* R613. Tested against the same strains of wild-type, the PURs-based NPs exhibited better antibacterial activity on the Gram-positive *L. monocytogenes* than on the Gram-negative *S. minnesota* and excellent activity against *S. Minnesota* R613 [167].

#### 4.3.3. Solid NPs to Exploit the Antimicrobial Properties of Essential Oils (EOs)

To counteract foodborne pathogens, which are the cause of endemics diseases worldwide, and limit the alarming problem of the increasing antibiotic drug resistance, BACs that have shown antibacterial activity in in vitro tests could be promising safe solutions. This is the case with EOs and of their constituents, but their intrinsic low stability, water-insolubility, and poor bioavailability limit their effectiveness in vivo. With the aim of ameliorating their performances, several EOs and their constituents have been subjected to modifications by nanotechnology and converted into NPs formulations for improving their antimicrobial activity, thus allowing their exploitation to extend food shelf-life and to minimize the growth of foodborne pathogens. Some examples of nano-formulations of EOs and of EOs constituents are reported in Table 12.

## 5. NPs Applications in Food: A Plethora of Advantages vs. Possible Risks and Limited Knowledge

It is evident that the application of nanotechnology for formulating BACs such as NPs with healthy purposes can allow us to achieve many nonpareil advantages, including the possibility to efficiently administer BACs that otherwise not soluble and not bioavailable and to exploit their beneficial and precious properties. Furthermore, the nanotechnological manipulation of BACs provides the possibility to process the unstable ones to enrich food, thus achieving FFs, to prepare FSs, and to manufacture active FP or preservative additives. Not only the specific food sector can take advantage of NPs applications, but so can the FP, cosmetics, and natural drugs sectors, while also attaining better acceptance of costumers.

Presently, many applications of nanotechnology to BACs are carried out to develop food-related unconventional therapeutics, food additives, and active/smart FP. Concerning FP, although many nanocomposites are still being researched and are still at the lab stage, several products are already allowed on the market by EFSA and by the Member States and the European Commission [168]. The application of nanotechnology to food and beverages has grown dramatically over the past 15 years, and due to the valuable properties, which nanomaterials can provide, the availability of BACs and foods nanotechnologically manipulated in the current market is destined to increase further. However, the major problem remains to be solved concerning the poor knowledge concerning the possible migration of the NPs from FP into foodstuff and the possible hazardous effects on consumers’ health, which can derive from ingestion and exposure to NPs [169]. Indeed, the enlargement of the development of nanomaterial-based food-related products is a topic debated incessantly among researchers with contrasting opinions, thus spreading concern and prejudice both among producers in various sectors and among consumers [170].

### 5.1. About NPs Application in FP: The Possible Migration of NPs to Food

Using European and U.S. (FDA) standard migration tests, the migration rate of NPs from nanocomposites for FP into food or food simulants has been assessed, and limits have been established by the European Regulation (European Commission, 2011). Nevertheless, in addition, to the data concerning only inorganic NPs, numerous issues exist that complicate the determination and interpretation of NPs migration studies and related results [169].

However, the list of authorized substances for manufacturing polymeric food contact materials published by the European Commission [169] includes nano-clay, titanium nitride, nano-silver, silanated silicon dioxide, titanium oxide, zinc oxide, and iron oxide NPs. Concerning these NPs, the reported opinions have established that the migration of NPs from FP would be low and slow [168].

In this regard, by investigating the migration of nano-clay particles into vegetables packaged with starch/clay nanocomposite, it was established that the overall migration is in conformity with European directives [168]. A nanocomposite film made of PLA/laurate reinforced with layered double hydroxide (LDH-C12) was evaluated using a simulant of fatty food, for assessing the possible migration of NPs in meat, providing results below the legal migration limits [171]. In another study that used different food simulants, it was established that carbon nanotube (CNT) NPs, when amalgamated with low-density polyethylene (LDPE) and polystyrene (PS), do not migrate [172]. Furthermore, the NPs migration and the antibacterial properties of an already marketed FP containing AgNPs used to package chicken meatballs were evaluated under the domestic storage conditions [173]. Even if no antibacterial activities were observed, the silver NPs migration was detected but was encouragingly slow. On the contrary, no transfer of silver NPs into chicken breasts and distilled water from AgNPs/co-polyester films was detected in a study by Tiimob et al. [174]. Curiously, it was evidenced that organic additives, such as Irganox 1076, Irgafos 168, Chimassorb 944, Tinuvin 622, UV-531, and UV-P, improve the migration of silver NPs from nano-silver-polyethylene packaging films into acidic food simulants (3% acetic acid) [175].

### 5.2. About NPs Application in Food: The Possible Toxicity of Ingesting NPs

Another question to solve is the need to have an ever-wider knowledge on the effect that the NPs-based BACs, food, and additives could have on human health when ingested and on the environment if scattered around. The human body was not set up for eating and metabolizing NPs, and ironically, although the application of nanotechnology in food offers new opportunities to ingest higher-quality aliments, also possessing improved beneficial properties, it might also pose new risks to human health and the environment [169]. The health risk associated with the ingestion of food containing nanomaterials is not yet fully clear. Note that small particles are absorbed easier and faster, thus quickly reaching organs where they can accumulate and damage cells and tissues, by exerting both direct and indirect toxicity [176]. Paradoxically, in complex biological systems, NPs could generate pro-oxidants species, thus triggering uncontrolled productions of ROS, RNS, and free radicals, favoring OS, inflammation, and related pathological situations, including immune toxicity. Furthermore, residual solvents and polymers’ intrinsic toxicity could initiate immune reactions [168]. The toxicological consequences of assuming NPs mainly depend on their size, shape, surface charge, and their capacity and modalities to cross biological membranes [168]. Deleterious effects on liver, kidney, spleen, and possible allergic reactions have been observed after ingestion of foods containing BACs-loaded NPs, and PEG-grafted liposome infusions have caused non-IgE-mediated signs of hypersensitivity [168].

On the contrary, no significant cytotoxic effect or neurotoxic event was observed after the administration of different concentrations of free-anthocyanins and anthocyanins loaded in PLGA-based NPs [164]. Nevertheless, the risks to human health that could be associated with the long-term use of nanosized devices are to be more deeply explored [155].

Presently, we possess data concerning the toxicological profile of different kinds of inorganic NPs, which have been evaluated both in vitro and in vivo [177]. In vitro experiments were performed on human and/or rodent cell lines from the intestine, liver, lung, and skin and in plants cells. The types of assays carried out include the lactate dehydrogenase release assay, live-dead assay, cell counting, alamar blue assay, neutral red uptake, protein content, and trypan blue dye exclusion. The changes in different biomarkers, such as the production of ROS, levels of glutathione (GSH), inflammation responses, DNA damage, and cell death were considered for establishing the toxicity mechanisms. Furthermore, genotoxicity assays, including Comet assay, Ames test, and micronucleus assay have been conducted. In vivo experiments were done on rodents sub-chronically and chronically exposed to different kinds of inorganic NPs to evaluate macro-toxic or histopathological effects [177]. Unfortunately, the results from both in vitro and in vivo studies are frequently conflicting and debatable. Experiments on mice and rats have shown that multi-walled CNTs cause peritoneal mesothelioma [178]. Ames test on human umbilical vein endothelial cells (HUVEC) (0–125 mg/mL) showed that unmodified clay (Cloisite^®^ Na^+^, BYK-Chemie GmbH, Wesel, Germany) is not cytotoxic and mutagenic, while the organoclay (Cloisite^®^130B, BYK-Chemie GmbH, Wesel, Germany) (0–250 mg/mL) is detrimental [177]. From in vitro and in vivo studies, it was evidenced that TiO_2_NPs can accumulate in the tissues of mammals and are eliminated very slowly, but conclusions about the risk of oral ingestion of TiO_2_NPs are impossible because available toxicokinetic data are contradictory and not reliable [169]. Genotoxic and cytotoxic effects of AgNPs (2 and 8 nm) have been assessed on root meristematic cells of *Allium cepa*, observing a dose-dependent decrease in mitotic index and a chromosomal aberration number increase [179]. While studies exist reporting the cytotoxicity, ROS generation, and hazardous liver accumulation of SiO_2_NPs, other studies have established no accumulation or toxicity in rats [169]. In in vitro models, SiO_2_NPs was shown to damage the gastrointestinal apparatus and its function, by injuring the brush border membrane both acutely (4 h) and chronically (5 days) [180]. An ROS-dependent toxicity was assumed for SiNPs, FeONPs, and Fe_2_O_3_NPs, but no accumulation in tissues and no toxicity was detected after their oral administration in high doses over a period of 13-week to rats. It is reported that, through ingestion, inhalation, and parenteral routes, ZnONPs can reach several organs [181]. In particular, their oral administration induces neurotoxicity and proinflammatory response in rats and immune reactions in BALB/c mice [182,183]. Furthermore, recent in vitro experiments have demonstrated that ZnONPs may cause a decrement in the transport of Fe and glucose and affect the microvilli of the intestinal cells [184]. A reduction of the toxicity of ZnONPs was successfully realized by modifying NPs surfaces with a silica coating [168].

### 5.3. Authors’ Considerations

Collectively, the above-mentioned scenario has generated various considerations in us. It highlights that further studies on toxicity, ecotoxicity, migration tendency, and risks associated with the intake of NPs are indispensable to authorize a massive application of NPs in the food area. Concerning nanomaterial-improved FP, clarifications about the behavior of NPs once implanted in packaging and in contact with the different packaged foodstuff are necessary. More in-depth knowledge concerning the mechanisms involved in NPs migration and how the diffusion process can influence the size and morphology of NPs is required. More standardized food models (SFMs) such as that proposed by Zhang and co-authors for evaluating the impact of the food matrix on the toxicity and fate of NPs after ingestion should be designed and developed [185]. Additionally, toxicological evaluations for each type of NP should be performed, and the concentrations tested should consider the average amounts of NPs usually added in nanocomposites. More data from studies on an increasing number of nanocomposites are necessary for a more reliable evaluation of nanotoxicological aspects of BACs-loaded nanomaterials. Furthermore, researchers should consider more carefully the issues associated with BACs-NPs’ oral administration and the possible degradation processes, which can occur in the GIT lumen, thus destroying the BACs formulations before being absorbed. Even if in vitro results concerning the use of NPs for BACs administration are promising, to forecast in a reliable way their actual oral bioavailability, in vivo experiments also considering the impact of GIT digestion are required. To this end, in addition to implementing models mimicking the salivary, gastric, and salts and enzymes concentrations in the intestinal fluids, it is necessary to include also proper simulators of GIT dynamics, structure, and mechanical issues. Moreover, the gut metabolism can affect the stability of the NPs-based delivery systems and the biological activity of BACs-loaded NPs.

Often, NPs are designed mainly with the aim of improving the absorption of BACs at the gastric or small-intestine level, thus not considering that the healthy properties of some phytochemicals are mainly due to metabolites that are formed only in the large intestine for the metabolic action of local microbiota. In these cases, the absorption in the upper GIT is undesirable. Special attention should be paid to eventual modifications that can occur in the NPs structure, depending on the in vivo digestive conditions, such as dilution by GIT fluids, pH, presence of ions, and enzymatic activities. Additionally, by loading BACs in NPs, modification in cellular signaling routes can occur, and cells’ responses after BACs-loaded NPs oral administration could differ from the administration of the free forms of BACs.

## 6. Conclusions and Future Perspectives

In the coming years, research should focus on designing and developing new strategies for maximizing the efficiency of BACs-loaded NPs; therefore, different nanomaterials should be prepared with characteristics specifically suitable for the diverse types of BACs. It is evident that the interactions between BACs and nanocarriers or between them and molecules present in the complex biological systems need further in-depth investigations. A considerable number of topics about NPs-mediated BACs administration remain underexplored. Finally, despite the evident contribution of nanotechnology in enhancing the bioavailability and bioactivity of BACs, it must be considered that NPs are often synthesized by physical and chemical methods using high-cost and hazardous chemicals, which could limit their high-scale production and could have a considerable environmental impact in terms of the residues that derive from the NPs synthesis. Green technologies using no toxic reagents to prepare NPs are required, and the synthesis of NPs using eco-friendly and biocompatible reagents could be a promising solution and an appealing contribution to minimizing the side effect of these processes.

## Figures and Tables

**Figure 1 polymers-13-02262-f001:**
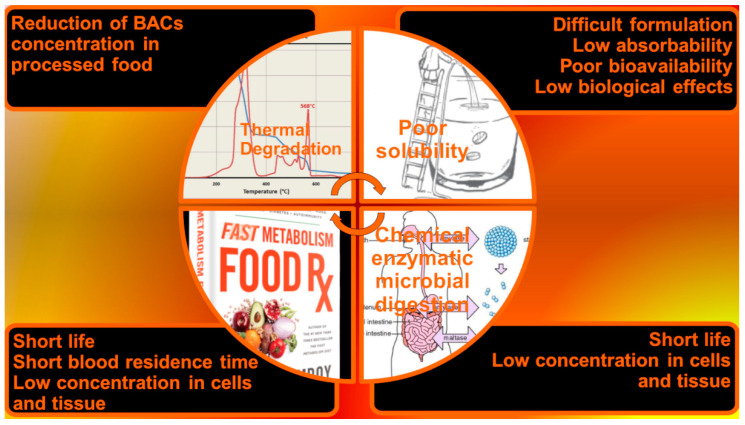
Main criticisms and events that limit the beneficial effects of BACs after oral administration.

**Figure 2 polymers-13-02262-f002:**
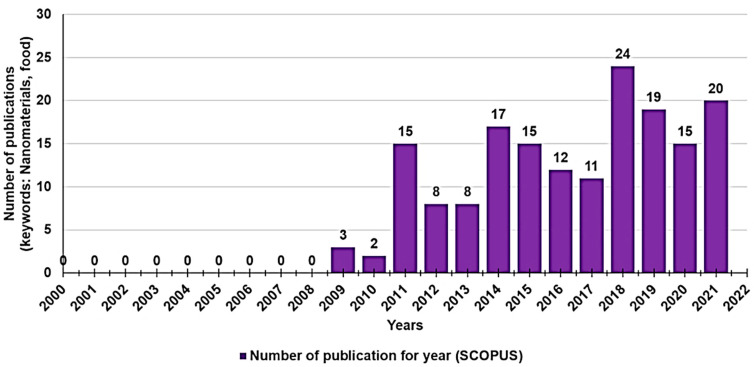
Number of publications per year of the last 20 years according to Scopus.

**Figure 3 polymers-13-02262-f003:**
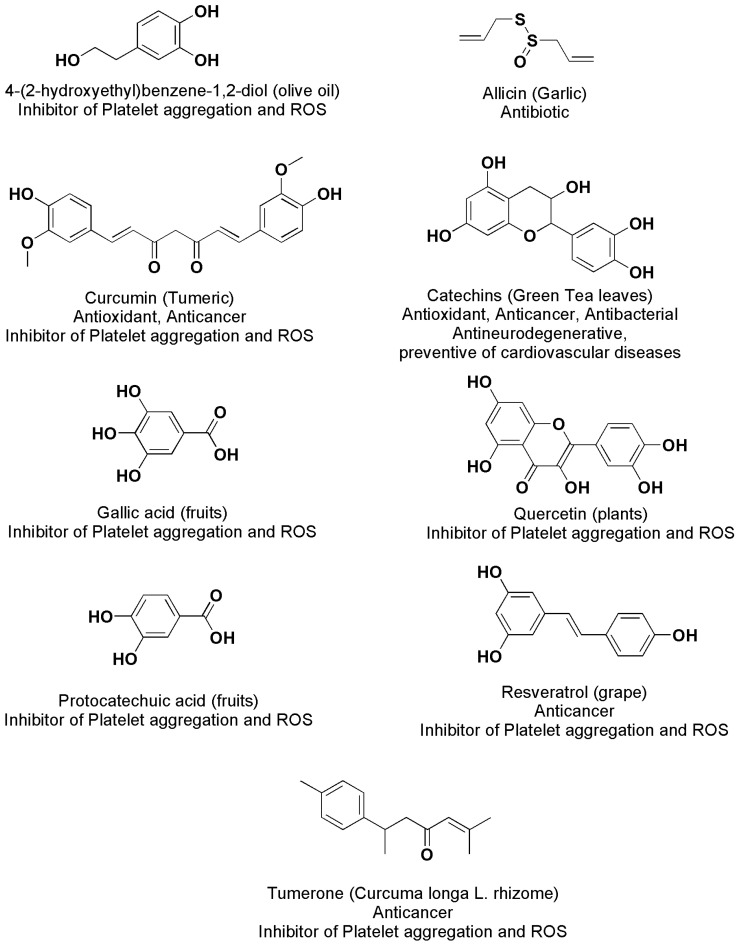
Structures of BACs mentioned in the main text.

**Figure 4 polymers-13-02262-f004:**
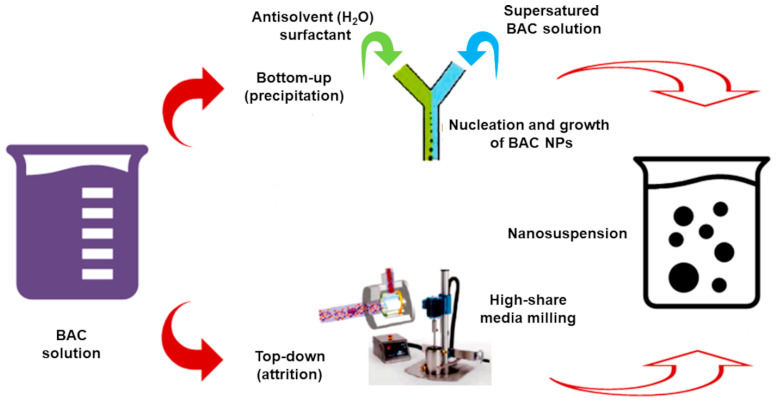
Bottom-up and top-down methods.

**Figure 5 polymers-13-02262-f005:**
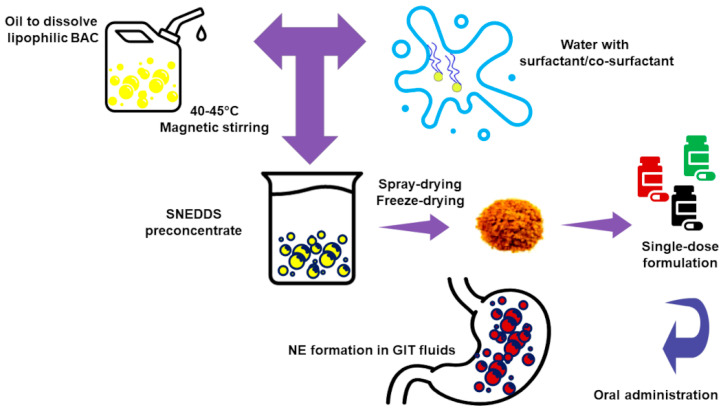
Schematic process for preparing SEEDSs and formation of NE in GIT fluids.

**Figure 6 polymers-13-02262-f006:**
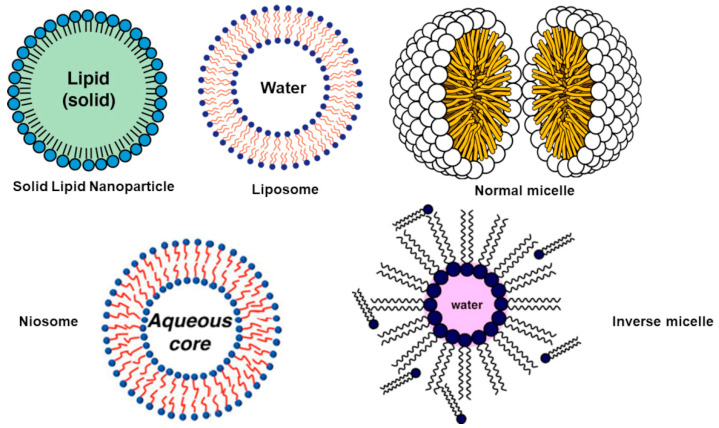
Examples of lipid-based NPs.

**Figure 7 polymers-13-02262-f007:**
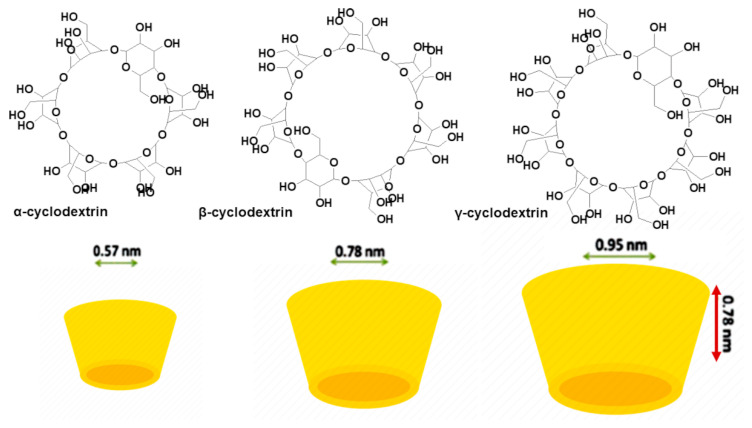
Chemical structure, spatial arrangement, and size of CDs.

**Figure 8 polymers-13-02262-f008:**
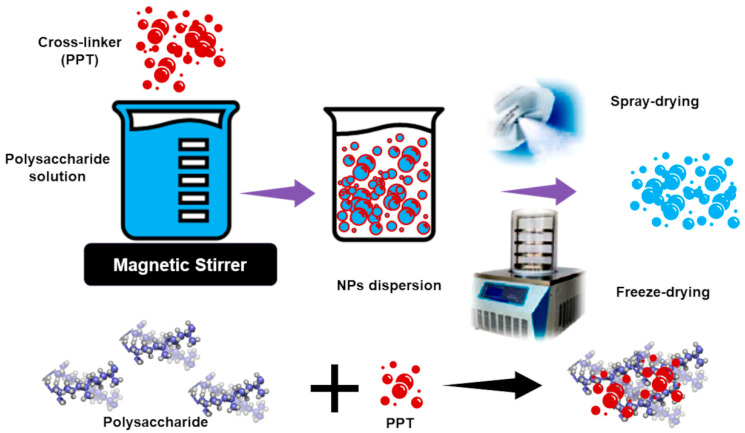
Process to prepare PNPs.

**Figure 9 polymers-13-02262-f009:**
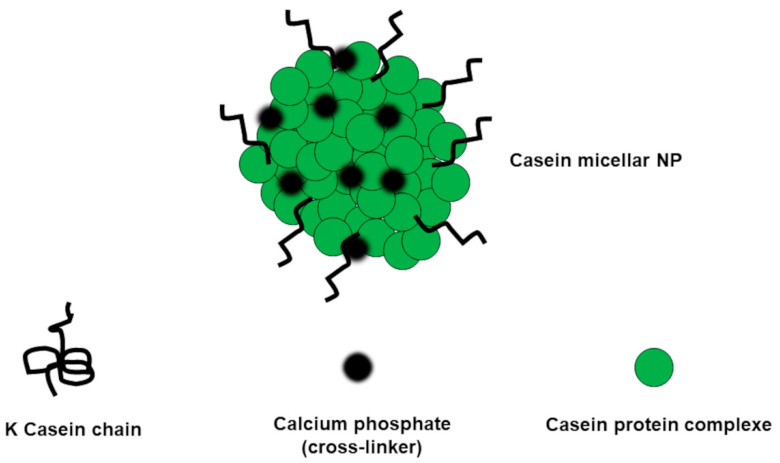
Schematic representation of a casein-based NP.

**Figure 10 polymers-13-02262-f010:**
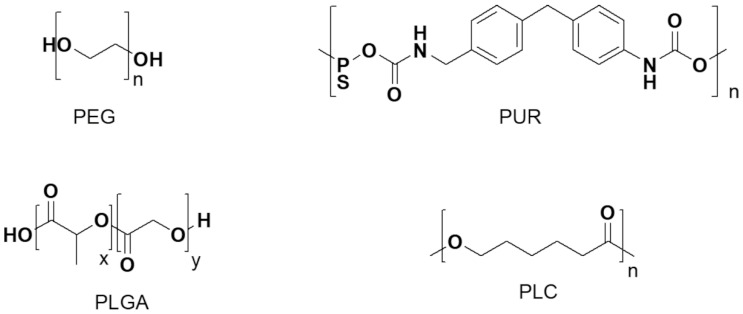
Some of the most used organic polymers adopted to produce nanosized GRAS delivery systems.

**Table 1 polymers-13-02262-t001:** The main families of BACs, their sources, nutritional functions and established healthy properties.

Chemical Category	Description	Source	Nutritional Function	Healthy Properties
Carotenoids	Plant pigmentsLipid nature	Carrots, pumpkins, melon Apricots, tomatoes Watermelons, peppers, Spinach, cabbage, parsley	Vitaminic activity(retinol)	Photoprotective agentsAntioxidantsImmune system reinforcers 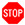 precancerous disease progress
Hydro-soluble vitamins (HVs)Liposoluble vitamins (LV)	Hydro-soluble/Liposoluble Organic Compounds	Fruits, vegetables, fish Meat, eggs, dairy products Milk	Health of cells, organ, tissuesPromote the use of thethe energy supplied by food	Prevention of many pathologiesTreatment of post-operative debilitation, Treatment of severe stress Bone reinforcement
Phytosterols	Saturated/not saturated sterols and stanols	Plants, cereal products Vegetables, fruit, berries Vegetable oils	N.P.	↧ LDL cholesterol
Polyunsaturated lipids	Unsaturated long chain fatty acids	Fatty fish, plant-based oils Seeds, nuts	Supply calories ↑ absorption of LVsProvide essential nutrients	Crucial for brain development and function ↧ Age-related mental decline
Curcuminoids	Linear diarylheptanoids	Turmeric, curry powder Mango, ginger	N.P.	Analgesic, anti-inflammatory Anti-cancer, antioxidative Anti-depressive Against hay fever, depression↧ Cholesterol and itching risk
Polyphenols	FlavonoidsTannic acidEllagitanninsPhenolic acids	Fruits, vegetables, grains bark, roots, stems, flowers Tea, wine	N.P.	Anti-oxidative, anti-inflammatory, Anti-mutagenic, anti-carcinogenic Modulate cellular enzyme functions
Indole compounds (indole-3-carbinol)	Organic compounds containing the indole structure	Cabbage, cauliflower Broccoli, kaleBrussels sprouts	N.P.	Strong antioxidant, DNA protectorChemo-preventive, anti-cancer↑ Heart health
Alkaloids	Basic organic compounds containing at least one nitrogen atom	Bacteria, fungi, plantsAnimals	N.P.	Antimalarial, antiasthma, anticancer Cholinomimetics, vasodilatory Antiarrhythmic, analgesic Antibacterial, antihyperglycemic Psychotropic and stimulant activities
Phytoprostanes Phytofurans	Oxylipins synthesized by the oxidation of α-linolenic acid	Almonds, vegetal oilsOlives, algae, passion fruit Nut kernels, rice	N.P.	Immunomodulators, anti-inflammatory Anti-tumors

N.P. = Not possessed; 
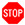
 = stop at; ↑= increasing; ↧ = decreasing.

**Table 2 polymers-13-02262-t002:** Main factors which influence the solubility of a substance.

Factor	Explanation	Reference
Particle size Shape	Unsymmetrical, ↧↧↧ small size particles with ↑ surface area dissolve better and more quickly	[42]
Temperature	↑ temperatures promote dissolution	[43]
Molecular Weight (MW)	↑ MW = ↧ its solubility	[44,45]
Chemical structure	↑ amount branching in carbon chains = ↑ solubility Branched polymers are ↑ soluble of linear ones with = MW Branched-chain molecules have ↧ volume/dimension ratio in solution and ↑ dissolution rate	[44,45]
Molecular Polarity	Polar solvents dissolve polar solutes Non-polar solvents dissolve non-polar substances	N.M.
Physical form	Molecules arranged in amorphous forms possess ↑ aqueous solubility than the crystalline ones Different polymorphs have ≠ solubility	[46]
pH of the medium	Weak acids and weak bases ionize in solution Ionized forms have ↑ solubility in water	N.M.
Presence of surface-active compounds with a hydrophilic head and a lipophilic tail	↧ the interfacial tension between the oil and water interface = ↑ both phases solubility	[15]

N.M. = not mentioned; ↧↧↧ = strong decrement; ↑ = increasing; ↧ = decreasing.

**Table 3 polymers-13-02262-t003:** Techniques and approaches for enhancing the solubility of BACs.

**Techniques**		**Goal**	**Strategies**	**Methods**	**Reference**
	PETs	BACs with modified physicochemical properties	↧↧↧↧ particles size No stabilizers No surfactants	Mechanical particle-size reduction Wet-milling, dry-milling High-pressure homogenization Ultra-high-pressure homogenization	[47,48]
				Cryogenic particle engineering Nanoprecipitation Nanosuspension Supercritical fluid processing	[47,49]
				Freeze-drying, spray-freezing	
	FAs	Solid formulations Lipid formulations	Use of mixture of water/oil phases, stabilizers, solvents/co-solvents	Spray-drying Milling	[15]

↧↧↧↧ = very strong decrement.

**Table 4 polymers-13-02262-t004:** Details concerning the most used top-down and bottom-up methods to prepare NSs.

Approach	Technique Type	Instruments	Advantages	Limitations
		Solvents		
		Additives/Polymers		
Bottom-up	Nanoprecipitation (solvent/anti-solvent) [64]	Mixer (pre-grinding)	Nanosized particles (Ps) Simple ↓ Cost equipment	Only for BACs soluble in organic solvents Residual organic solvents
		Organic solvent Water (anti-solvent)		
		Surfactants		
Bottom-up	Supercritical fluid extraction (SFE)	Syringe or diaphragm CO_2_ pumps	Nanosized Ps ↑ Selectivity ↑ Speedy	↑ Costs
		Solvents/co-solvents		
		No		
Bottom-up	Inclusion complexation (IC) [65] ^1^	No	Nanosized Ps Masking of odors/flavors Aroma’s preservation ↑ EE% ↑ Stability	For few materials
		Organic solvent, water		
		β-cyclodextrin β-lactogloglobulin		
Bottom-up	Coacervation [65]	No	Nanosized Ps ↑↑↑ payloads (99%) Controlled release Sustained release	↑↑↑ Influencing variables
		Organic/aqueous solvents		
		Polymers Chemicals/enzymatic cross-linkers (glutaraldehyde, transglutaminase) [62]		
Top-down (dissocubes)	HPH at r.t. [15]	Piston-gap homogenizers	Nanosized Ps No material erosion For solving both organic and aqueous solubility drawbacks	Pre-processing micronization Thermic degradation ↑ Cost instruments
		Aqueous media		
		Surfactants		
Top-down (Nanopure^®^)	Deep-freeze Homogenization	Piston-gap homogenizers	Nanosized Ps No crystals grow ↓ Operation times ↑ Stability	Pre-micronization ↑ Cost instruments
		Non aqueous media Water with water-miscible liquids		
		PEG-400, PEG-1000		
Top-down (IDD-P) ^2^ [15]	Jet stream Homogenization [66]	Z-type or Y-type Collision chamber	Nanosized Ps No crystals grow ↓ Operation times ↑ Stability	Pre-micronization Thermic degradation ↑ Cost instruments
		Aqueous media		
		Phospholipids, surfactants, stabilizers		
Top-down	Media milling technique collision [67]	High-shear media mills/Pearl mills	Nanosized Ps ↓ Batch-to-batch variation	↑ Operation times Thermic degradation
		Aqueous media		
		Excipients		

^1^ hydrogen bonding, van der Waals force or an entropy-driven hydrophobic effect; ^2^ Insoluble Drug Delivery-Particles; ↑↑↑ = strong increment; ↑ = increasing; ↧ = decreasing.

**Table 6 polymers-13-02262-t006:** Emulsion-based delivery systems for delivering BACs.

BACs	Method	Surfactant(s)	Results	Reference
Carotenoids (Paprika Oleoresin)	Solid self-microemulsifying carotenoid system (S-SMECS)	Tween 80	↑ Solubility	[85]
Lutein	SMEDDS	Tween 80 Labrasol TranscutolHP/Lutro-E400 ^1^	↑ Solubility ↑ Bioavailability	[86,87]
Polymethoxyflavones (PMFs)	HPH (NEs-based)	Tween 20 Tween 85	↑ Dissolution rate	[88]
*β*-Carotene	HPH (o/w NEs-based)	Tween 20	↑ Emulsion stability ↑ Solubility ↑ Bioaccessibility	[89]
Lycopene	MEs-based method	Ethoxylated sorbitan esters 3GIO SML	↑ Solubility	[90]
Quercetin	SNEDDS	Tween 80 PEG 400	↑ Solubility	[91]

^1^ co-surfactant; 3GIO = tri-glycerol monooleate; SML = sucrose monooleate; PEG = polyethylene glycol; ↑ = increasing.

**Table 7 polymers-13-02262-t007:** The main features that the SNDSs for food uses should possess.

SNDSs		
Requisite	Description	Refs.
Food grade ingredients	Manufactured with food-grade/natural ingredients No use of solvents	[15]
Food incorporation	Able to physically incorporate or covalently bind BACs Able to compact BACs in more soluble and stable NPs suitable for being incorporated into the food matrix with ↑ EE% and ↓ impact on the sensory properties of the derivative product	[100]
Protection by degradation	Able to protect BACs from the interaction with the food matrix constituents, temperature, light, pH, during food manufacturing, storage, processing, and from inactivation by digestion	[15]
↑ Uptake ↑ Bioavailability	Able to promote the cell up-take Able to release BACs in a controlled mode responding to specific environmental stimuli.	[15]
Industrial scalability	Suitable to be produced on a large scale	[101]

↑ = increasing; ↓ = decreasing.

**Table 8 polymers-13-02262-t008:** Examples of nanosized formulations of BACs developed by using the abovementioned polymers.

Polymers	BAC	Formulation	Properties	Reference
Modified PVA	ATRA	Micelles	↧ ATRA release ↥ Cytotoxic activity on NB cells	[107,108]
TPGS	ATRA	Micelles	↥ Cytotoxic activity on NB cells	[109]
Hydrophilic and amphiphilic biodegradable dendrimers	Ursolic acid Oleanolic acid	Dendrimer NPs	↥ Water solubility ↥ Biocompatibility ↥ Biodegradability ↧ Toxicity ↥ Antibacterial activity	[110,111]
	EA		↥ Water solubility ↥ Biocompatibility ↥ Biodegradability ↧ Toxicity ↥ Antioxidant properties ↥ Scavenging activity	[16]
	GA (linked)		↥ Solubility in lipids ↥ Biocompatibility ↥ Biodegradability ↥ Antioxidant properties ↥ Scavenging activity ↧ Platelet aggregation ↧ ROS production	[112,113,114]
	GA (Linked and encapsulated)		↥ Solubility in lipids ↥ Biocompatibility ↥ Biodegradability ↧ Toxicity ↥ Antioxidant	[112]

↑ = increasing; ↓ = decreasing.

**Table 9 polymers-13-02262-t009:** Main properties and functions of organic SNPs.

Nanomaterial	Requisites	Function	Reference
LNPs	↥ Functional groups	Host/guest interaction with HBACs and LBACs	[103,116]
	Inner cavities	Accommodation of LBACs	[103]
	Nanocontainers Protective envelopes	Opposite GIT digestion Opposite degradation	[117,118,119]
	Permeability enhancers	Promote GIT absorption	[117,118,119]
Polymeric SNPs	↥ MW	↥ Systemic retention time	[120]

↑ = increasing.

**Table 10 polymers-13-02262-t010:** Most common methods for preparing CD-ICPXs.

Method	Mixing	Solvent	Drying	Instruments	Steps
	Physical State				
Physical blending	Mechanical	No	No	Mixer	Mixing
	Powder				
Kneading	Mechanical	Water Water/alcohol	Yes	Kneading machine	CDs pasting BAC addition Mixing Drying
	Paste				
Co-precipitation	Mechanical	Solvent (BAC) Water (CD)	Yes	Magnetic stirrer Mechanical stirrer	Dissolutions Precipitation Drying
	Solutions				
Ball milling	Mechanical	No	No	Mechanical Oscillatory mill	Mixing
	Solid state				
SD ^1^	N.R.	Solvent (BAC) Water (CD)	No	Spray dryer	Dissolution SD
	Solutions				
FD ^2^	N.R.	Solvent (BAC) Water (CD)	No	Freeze dryer	Dissolution FD
	Solution				
Supercritical anti-solvent	Mechanical	Solvent (BAC) Water (CD) CO_2_ (anti-solvent)	No	Magnetic stirrer Mechanical stirrer	Dissolution Precipitation Solvent extraction
	Solutions/gas CO_2_				

N.R. = Not reported; ^1^ non-suitable for thermosensitive compounds; ^2^ suitable for thermosensitive compounds.

**Table 11 polymers-13-02262-t011:** Main polyphenols-enriched nanomaterials obtained using CDs.

Cyclodextrin	BACs	Improvements
β-CDs	Linoleic acid (LA)	↥ Thermal stability, ↧ Degradation
β-CDs	RES	↥ Stability, ↥ Solubility
Maltosyl-β-CDs		
Hydroxypropyl-β-CD	Carotenoids	↥ Water solubility
β-CDs	Lycopene (Lyc)	↥ Water dispersibility, ↥ Stability
HP-β-CDs	Hesperidin	↥ Stability, ↥ Solubility
β-CDs	Olive leaf extracts ^1^	↥ Water solubility, ↥ Stability, ↥ Antioxidant activity
HP- β-CDs Maltosyl-β-CDs β-CDs	Quercetin Myricetin	↥ Water solubility, ↥ Stability, ↥ Antioxidant activity
HP-β-CDs	Kaempferol	↥ Water solubility, ↥ Stability, ↥ Antioxidant activity
α-CDs β-CDs	3-Hydroxyflavone Morin Quercetin	↥ Water solubility, ↥ Stability, ↥ Antioxidant activity
β-CD	Rutin	↥ Water solubility, ↥ Stability, ↥ Antioxidant activity
HP-β-CDs	Curcumin	↥ Water solubility, ↥ Stability, ↥ Antioxidant activity
α-CDs	Ferulic acid	↥ Water solubility, ↥ Stability, ↥ Antioxidant activity
β-CDs	EA	↥ Water solubility, ↥ Anti-inflammatory activity
β-CDs/DMC	EA	↥ Water solubility, ↥ DL%, Controlled release, ↥ Oral bioavailability
α-CDs	Amino acids Hydrolysed soy pro ^2^	↧ Bitter taste perception

^1^ Rich in oleuropein; ^2^ proteins; ↑ = increasing; ↓ = decreasing.

**Table 12 polymers-13-02262-t012:** Examples of associations of essential oils (EOs) and of EOs constituents with polymeric NPs.

Formulation	BAC	Targeted Microorganism	Activity
PLC-based NPs	Tea tree oil	*Tricophyton rubrum*	↑ EO effectiveness against fungi infecting nails
Zein-Sodium Caseinate NPs	Thymol	*E. coli*, *Salmonella*	Good antimicrobial activity Two-phase release
PLGA-based NPs	Cinnamaldehyde Eugenol	*Salmonella* spp. *Listeria* spp.	Good antimicrobial activity Controlled release
Chitosan-based NPs	Carvacrol	*E. coli*, *S aureus*, *B. cereus*	↑ Antimicrobial activity
Chitosan-based NPs	Oregano EO	N.R.	Controlled release
Zein-based NPs	Thymol Carvacrol	*E. coli*	Good antimicrobial activity ↑ Solubility
Chitosan-based NPs	Eugenol Carvacrol	*S. aureus*, *E. coli*	↓ Cytotoxicity
PLGA-based NPs	Carvacrol	*S. epidermidis* biofilms	↓ Elasticity and stability of performed biofilm
Methyl/ethylcellulose NPs	Thymol	*S. aureus*, *E. coli* *P. aeruginosa*	Good antimicrobial activity Preventive activity in cosmetic lotions, creams, gels

N.R. = Not reported; ↑ = increasing; ↓ = decreasing.

## Data Availability

Not applicable.

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
