# Peer review of "Nanotechnological Manipulation of Nutraceuticals and Phytochemicals for Healthy Purposes: Established Advantages vs. Still Undefined Risks"

_polymers, 2021, doi:10.3390/polym13142262_

Round 1
Reviewer 1 Report
The authors proposed a review of about the nano-technological manipulation based on nutraceuticals and phytochemicals.
The manuscript is well drafted and structured with inherent logical connection between sentences and sections. English language require minor verification.
Author Response
We thank the Reviewer for his positive general comments concerning our study and for his suggestion.
As requested by the Reviewer, the manuscript has been carefully re-read by all authors to eliminate coarse errors and subsequently by Professor Deirdre Kantz, English mother tongue, working at the University of Genoa and Pavia, to further refine the quality of the language. Consequently, Prof. Kantz has been thanked in the proper section (lines 1187-1188). Corrections have not been highlighted in the text.
Reviewer 2 Report
The review "Nanotechnological Manipulation of Nutraceuticals and Phytochemicals for Healthy Purposes: Established Advantages vs Still Not Defined Risks" is comprehensive and contains the latest update. The information provided is of interest to researchers in this field. The authors discuss both views about the use of nanoparticles in the food area. The title of the paper clearly reflects its contents. There are just some minor issues to be addressed:
Why are flavonoids and polyphenols in Table 1 different chemical categories? Flavonoids are a family of polyphenolic compounds.
Line 147 and Figure 3… .For the bioactive constituent from olive oil different names are used. It would be better that the same name is used and that the name most represented in the literature is mentioned.
Page numbering is incorrect.
Table 2 should be explained in the text.
Line 818…Please replace full stop with a colon.
Author Response
The review "Nanotechnological Manipulation of Nutraceuticals and Phytochemicals for Healthy Purposes: Established Advantages vs Still Not Defined Risks" is comprehensive and contains the latest update. The information provided is of interest to researchers in this field. The authors discuss both views about the use of nanoparticles in the food area. The title of the paper clearly reflects its contents. There are just some minor issues to be addressed:
Why are flavonoids and polyphenols in Table 1 different chemical categories? Flavonoids are a family of polyphenolic compounds.
We agree perfectly with what the Reviewer noted. Indeed, flavonoids are a sub-category of polyphenols. Then, the flavonoids row was deleted, and the voice “flavonoids” was added in column 2 of the polyphenol row.
Line 147 and Figure 3… .For the bioactive constituent from olive oil different names are used. It would be better that the same name is used and that the name most represented in the literature is mentioned.
We thank the Reviewer for pointing this out to us and agree with his suggestion to standardize the two names used for the bioactive constituent of olive oil. We therefore decided to use the name shown in Figure 3 (4-(2-Hydroxy-ethyl)-benzene-1,2-diol), as it is chemically correct, according to IUPAC terminology. Please, see lines 147-148.
Page numbering is incorrect.
The Reviewer is right, and we are aware of it. But it is a drawback that happens when, to change the page layout, it is necessary to make a section break, and the insertion of the page numbers is done automatically by the template provided by Polymers. It is not possible to change them. For example, it is not even possible to use the word function "delete page numbers". We therefore ask the Reviewer to overlook this defect in the manuscript which (from experience) will be eliminated by the Editorial Office.
Table 2 should be explained in the text.
As requested, Table 2 has been explained in the text. Please, see lines 193-195.
Line 818…Please replace full stop with a colon.
As requested, full stop has been replaced with a colon. Please, see line 828.
Collectively, we thank the Reviewer for his positive general comments concerning our study and for his suggestions.
Reviewer 3 Report
This is a very complete review article regarding the nanotechnological transformations of bioactive constituents of plants, which aims to improve their stability, taste, appearance, quality,and lifetime. The authors discussed its biological properties and chemical structure and the aspects of nanotechnology and nanomaterials which could improve its effects. This is an important paper to the field, since it discuss several aspects of using nanotechnology for improving human health.
In the conclusion, the authors discuss the influence of gastrointestinal secretions in the nanoparticles. I guess this is not the conclusion of the study, it should be in a separate topic before the conclusion.
At figure 1, the excess of visual information turns it difficult to comprehend. I think it should be more clear, removing the background of the circle.
Author Response
This is a very complete review article regarding the nanotechnological transformations of bioactive constituents of plants, which aims to improve their stability, taste, appearance, quality, and lifetime. The authors discussed its biological properties and chemical structure and the aspects of nanotechnology and nanomaterials which could improve its effects. This is an important paper to the field, since it discuss several aspects of using nanotechnology for improving human health.
We thank the Reviewer for his positive general comments concerning our study and for his suggestions.
In the conclusion, the authors discuss the influence of gastrointestinal secretions in the nanoparticles. I guess this is not the conclusion of the study, it should be in a separate topic before the conclusion.
We appreciate the Reviewer suggestion. So, the part concerning the influence of gastrointestinal secretions in the nanoparticles has been removed (lines 1153-1172) from the conclusions section and it was moved in the previous paragraph (5.3. Authors Considerations). Please, see lines 1115-1116 and 1128-1148.
At figure 1, the excess of visual information turns it difficult to comprehend. I think it should be more clear, removing the background of the circle.
We agree. We have removed the background of the circle, as requested. In addition, we have made further changes, to ameliorate the image. Now the Figure 1 appears clearer and more comprehensible. We hope the Reviewer appreciate our modifications.